

# Local and remote temperature response of regional SO₂ emissions

Anna Lewinschal[1,2], Annica M. L. Ekman[1,2], Hans-Christen Hansson[2,3], Maria Sand[4], Terje
K. Berntsen[4,5], and Joakim Langner[6]

[1]Department of Meteorology, Stockholm University, Stockholm, Sweden
[2]The Bolin Centre for climate research, Stockholm University, Stockholm, Sweden
[3]Department of Environmental Science and Analytical Chemistry, Stockholm University, Stockholm, Sweden
[4]CICERO Center for International Climate and Environmental Research, Oslo, Norway
[5]University of Oslo, Department of Geosciences, Oslo, Norway
[6]Swedish Meteorological and Hydrological Institute, Air Quality Research Unit, Norrköping, Sweden

**Correspondence:** Anna Lewinschal (anna@misu.su.se)

**Abstract.** Short-lived anthropogenic climate forcers, such as sulphate aerosols, affect both climate and air quality. Despite
being short-lived, these forcers do not affect temperatures only locally; regions far away from the emission sources are also
affected. Climate metrics are often used e.g. in a policy context to compare the climate impact of different anthropogenic
forcing agents. These metrics typically relate a forcing change in a certain region with a temperature change in another region
and thus often require a separate model to convert emission changes to radiative forcing changes.
In this study, we used a coupled Earth System Model (NorESM) to calculate emission-to-temperature-response metrics for
sulphur dioxide (SO₂) emission changes in four different policy-relevant regions: Europe, North America, East Asia and South
Asia. We first increased the SO₂ emissions in each individual region by an amount giving approximately the same global
average radiative forcing change (-0.45 Wm$^{-2}$). The global mean temperature change per unit sulphur emission compared to
the control experiment was independent of emission region and equal to ∼0.006K/TgSyr$^{-1}$. On a regional scale, the Arctic
showed the largest temperature response in all experiments. The second largest temperature change occurred in the region of
the imposed emission increase, except when South Asian emissions were changed; in this experiment, the temperature response
was approximately the same in South Asia and East Asia. We also examined the non-linearity of the temperature response by
removing all anthropogenic SO2 emissions over Europe in one experiment. In this case, the temperature response (both global
and regional) was twice of that in the corresponding experiment with a European emission increase. This nonlinearity in the
temperature response is one of many uncertainties associated with the use of simplified climate metrics.
*Copyright statement.* TEXT

# 1  Introduction

Anthropogenic emissions of short-lived climate forcers (SLCFs), i.e. chemical components in the atmosphere that interact
with radiation, have both an immediate effect on local air quality and regional and global effects on the climate in terms of





e.g. changes in the temperature and precipitation distribution. Aerosol particles are one of the most important SLCFs due to their abundance and their effects on health and climate. The short atmospheric residence times of SLCFs such as sulphate and carbonaceous aerosols (around days) lead to high atmospheric concentrations in emission regions and a highly variable radiative forcing pattern. Regional radiative forcing can, nevertheless, exert a large influence on the temperature field away from the forcing region through changes in heat transport or the atmospheric or ocean circulation (Menon et al., 2002; Shindell et al., 2010; Lewinschal et al., 2013; Acosta Navarro et al., 2016; Dong et al., 2016). Here, we investigate the effect of sulphate aerosol precursor emission perturbations in different regions on the global surface temperature distribution using a global climate model.

The local radiative forcing by a unit aerosol emission varies from region to region depending on a number of factors, including e.g. emission location, aerosol processing in the atmosphere and removal rates as well as land surface properties and cloud distribution (e.g. Bellouin et al., 2016). Moreover, a unit radiative forcing in a specific region may have different impacts on the temperature response locally in the forcing region and in remote regions away from the forcing, as well as between different remote regions. In other words, the climate sensitivity in one region can vary depending on the location of the forcing (e.g. Shindell and Faluvegi, 2009).

To facilitate comparisons of the climate effect of different greenhouse gases and emission levels, several climate metrics have been developed which connect emission changes to radiative forcing, or a specified forcing to a temperature response (e.g. Aamaas et al., 2013). One appeal of simple climate metrics is that they provide a way to easily evaluate the climate impact of different air quality or climate mitigation policies without having to run a coupled climate model, something which is not always feasible due to the computational costs. Because of the even spatial distribution of long lived greenhouse gases, these metrics have usually described global average quantities. However, the highly variable spatial distribution of aerosol forcing necessitates the use of metrics that take these spatial inhomogeneities into account (Shine et al., 2005).

Shindell and Faluvegi (2009) developed a metric that accounts for spatial inhomogeneities both in the forcing and tempera-ture response, the Regional Temperature Potential (RTP). With a large set of simulations with one climate model, where they varied the location of forcing from various anthropogenic climate forcers, these authors derived RTP coefficients that link the radiative forcing from a climate forcer in a specific region to regional temperature responses. An evaluation of the method for transient simulations of historical aerosol forcing and response with four different climate models was presented in the work of Shindell (2012).

However, the simplification inherent in the climate metric concept might lead to difficulties related to the generality of these metrics, such as the RTP. Differences between RTP coefficients derived from different climate models can stem from a number of different sources, involving everything from atmospheric processing of aerosols, interaction with radiation, aerosol cloud effects or climate feedbacks, and how these processes are represented in different climate models (Kasoar et al., 2016).

The main objective of this study is to investigate the global and remote impacts of regional sulphate aerosol precursor emission changes on the surface temperature distribution. This is done by using a coupled atmosphere-ocean model with interactive aerosol representation, the Norwegian Earth System Model (NorESM). The results from the model simulations are used to derive RTP coefficients similar to the work of Shindell and Faluvegi (2009). However, our method for deriving RTP



coefficients differs from that of Shindell and Faluvegi (2009) in that we derive our RTP coefficients directly from emission
perturbations and focus primarily on the emissions-temperature connection rather than the connection between radiative forcing
and temperature. The RTP coefficients derived by Shindell and Faluvegi (2009) describe the regional temperature change in
response to regional radiative forcing, and essentially describe a regional sensitivity. These forcing-based sensitivities have to
be combined with the radiative forcing patterns derived from emission scenarios with a chemistry transport model or offline
calculations for radiative forcing with a general circulation model to provide the emission-temperature connection. Another
difference is that we focus on emissions from air-pollution and policy-making relevant regions rather than the latitudinal bands
of Shindell and Faluvegi (2009). Thus, we seek to investigate how much an emission change in one policy relevant region
affects both local climate as well as the climate on global scale and in remote regions.

The aim is that the RTP coefficients derived with NorESM eventually could be used in Integrated Assessment analysis

(IAA), e.g. such as the Greenhouse gas - Air pollution Interactions and Synergies (GAINS) model. In the GAINS model the
climate impact is estimated using the Global Warming Potential (GWP), which is the global radiative forcing integrated over
time normalised by that of $CO_2$ (Amann et al., 2011). By the GWP the global climate impact of SLCFs can be taken into
account. Lately the radiative forcing of long-lived greenhouse gas emission changes due to air pollution abatement, other than
$CO_2$, has been included. Using RTP coefficients in IAA would mean that not only near-term climate effects of changed SLCF
emissions can be evaluated but also how different regions are affected due to specific regional abatement measures. The RTP
can be based on different entities as radiative forcing, effective radiative forcing or direct emissions, which need very different
support calculations respectively. Using the emissions as base for RTPs will provide a very simple way to estimate the climate
impact of changed emissions without having to run a chemical transport model. Using any of the bases for the RTPs avoids
running large coupled climate models. However, the validity of this method relies on the accuracy of the assumption that the
temperature response to changed emissions is linear and that the interaction between different SLCF are negligible for the
resulting temperature response. To address the question regarding linearity in the response depending on emission perturbation
strength we perform simulations with different emission perturbations for the European region.

The layout of this study is as follows. First an introduction to the RTP methodology is presented in the method section. The

NorESM model is described together with the experimental design to derive the emission specific RTP coefficients. In Sect.
3 we first present the results from experiments where sulphate aerosol precursor emissions were increased and the global and
regional effect of these emission perturbations. The results of an experiment where European anthropogenic sulphate aerosol
precursor emissions were removed are discussed in the context of non-linearities emerging as a consequence of emission mag-
nitudes. Last in the Result section is an comparison of the performance of the forcing-based RTP coefficients of Shindell and
Faluvegi (2009) and Shindell et al. (2012) for NorESM results. The Result section is followed by a discussion and conclusions.





## 2 Method

### 2.1 The Absolute Regional Temperature Potential

There exists a number of different climate metrics that describe the connection between emissions of atmospheric tracer species
and/or their radiative forcing and/or their effect on the global mean temperature. Many have been developed for the purpose
of evaluating the impact of increased emissions of long-lived and well-mixed greenhouse gases. Thus, the connection between
the location of an emission perturbation and the temperature response has not been a primary concern. However, for SLCFs
the location of the emission perturbation and radiative forcing is a primary matter of interest. A climate metric which takes the
spatial distribution of these SLCFs and the temperature response into account was developed by Shindell and Faluvegi (2009)
and Shindell and Faluvegi (2010). The metric describes the temperature change $dT$ in one area $a$ at time $t$, in response to
forcing $F$ in area $a'$:

$$dT_a(t) = \int_0^t \left( \sum_{a'} F_{a'}(t') \cdot \frac{dT_a/F_{a'}}{dT_a/F_{global}} \right) \cdot IRF(t-t')dt', \qquad (1)$$

where the numerator in the second term of the sum, $dT_a/F_{a'}$, is the regional response coefficient (cf. Table 3 of Shindell
and Faluvegi (2010)), which, in this formulation is normalised by the regional temperature response to global average forcing,
$dT_a/F_{global}$. The Impulse Response Function, $IRF$, represents the time dependent temperature response per unit forcing, i.e.
the climate sensitivity. For the equilibrium (or quasi-equilibrium or transient) temperature response to a steady forcing, the
$IRF$ can be replaced by the equilibrium or transient climate sensitivity, $\lambda$.
Shindell (2012) elaborated the regional temperature change metric of Shindell and Faluvegi (2010) to an Absolute Regional
Temperature potential, $ARTP$, which, in analogue to the Absolute Global Temperature change Potential (AGTP), connects an
emission perturbation, $E$, in region $r$ of a climate forcer to an absolute temperature change (Shine et al., 2005) in area $a$:

$$ARTP_{a,r}(t) = \int_0^t \left( \sum_{a'} \frac{F_{a'}(t')}{E_r} \cdot \frac{dT_a/F_{a'}}{dT_{global}(F_{global})/F_{global}} \right) \cdot IRF(t-t')dt'. \qquad (2)$$

This formulation uses the global climate sensitivity ($dT_{global}(F_{global})/F_{global}$) to normalise the regional response coeffi-
cients in contrast to Eq. 1 which uses the regional sensitivity to global forcing. This, i.e. the second term in the summation of
Eq. 2, yields the unitless RTP coefficients presented in Table 1 of Shindell (2012). Shindell (2012) also advocate the use of the
latter formulation (Eq. 2) before the former (Eq. 1).
The RTP coefficients provided in the work of Shindell and Faluvegi (2010) and Shindell (2012) were derived for forcing in
four latitude bands covering the globe: the Southern Hemisphere extratropics (90-28°S, SHext), Tropics (28°S-28°N), Northern
Hemisphere mid latitudes (28-60°N, NHml) and Arctic (60-90°N). These RTP coefficients can be used to estimate the global
temperature response to any emission perturbation, as long as the forcing in response to the emission perturbation in each of



the latitude bands described above is known. The forcing distribution in response to an emission perturbation can be calculated
with e.g. a chemistry transport model (direct radiative forcing only), or with atmospheric general circulation models.

In this work, we take our starting point in emission perturbations rather than in the forcing distribution. Sub-global tem-

perature changes in response to emission perturbations are derived both for latitudinal bands following Shindell and Faluvegi
(2009) as well as for the emission regions defined in this study, with the addition of a complementary Arctic region (AR).
This complementary Arctic region is defined as the area north of the Arctic circle (66°N), whereas the northernmost latitudinal
band (hereafter denoted ARCT) is defined as the area north of 60°N in accordance with the definition of Shindell and Faluvegi
(2009). All regions that are used in this study are listed in Table 1.

## 2.2   NorESM

The regional temperature changes in response to aerosol emission perturbations are investigated using NorESM (Bentsen
et al., 2013). This model is based on the Community Climate System Model 4.0 (CCSM4.0), but has been modified to include
interactive aerosols and to use the Bergen version of the Miami Isopycnic Coordinate Ocean Model (MICOM) instead of the
Parallel Ocean Program (POP) model. For NorESM the atmospheric component of the model, the Community Atmospheric
Model version 4 (CAM4) has been extended with an interactive aerosol module, CAM4-Oslo (Kirkevåg et al., 2013). The land
surface is represented by the Community Land Model version 4 (CLM4) and sea-ice is modelled with the ice model CICE4.
The atmospheric model uses a finite volume grid with a resolution of 1.9°x2.5° latitude-longitude.

The aerosol module in NorESM considers five different aerosol components: sulphate, black carbon, organic matter, min-

eral dust and sea salt. Both the mass and number for these aerosol constituents are predicted in a combined sectional and
modal framework. Emissions take place both in the form of primary particles and as precursors to aerosols where the aerosol
chemical compounds are produced through aqueous and gas phase chemical reactions. Aerosols can exist both as external
and internal mixtures, depending on atmospheric processing. E.g. sulphate coating of black carbon, which changes the optical
and hygroscopic properties of this internally mixed aerosol compared with the externally mixed constituents, is accounted for.
Humidification of aerosols is based on the hygroscopicity of the aerosol and the atmospheric relative humidity. Aerosols are
removed from the atmosphere by dry and wet deposition.

Aerosol can affect cloud properties through acting as cloud condensation nuclei (CCN). The efficiency of a particular aerosol

depends on its hygroscopicity and size. The amount of aerosol particles that are efficient CCN is connected to the predicted
aerosol size and mass and connected to the two-moment cloud microphysics for stratiform clouds in the model. Thus, NorESM
simulates both the cloud albedo effect and cloud lifetime effects of aerosols. Beside these effects of aerosols on cloud mi-
crophysical properties, semidirect effects which depend on changes of the thermal structure of the atmosphere are accounted
for.

An evaluation of the performance of NorESM in simulating the present climate was carried out by Bentsen et al. (2013),

who identified the main biases in the modelled climate compared to observations and that the model simulates a stable climate.
Iversen et al. (2013) derived climate sensitivities for NorESM and investigated the climate response to different future emission





scenarios. They found that the $CO_2$ climate sensitivity of the model is smaller than the Coupled Model Intercomparison Project
phase 5 (CMIP5) multi-model mean, but within one standard deviation.

## 2.3 Experiments

We perform a suite of model simulations with NorESM where aerosol precursor emissions are perturbed in one region at a
time. Four regions which we consider being of particular interest from an aerosol and air-pollution perspective are studied:
Europe, North America, South Asia and East Asia. The emissions of anthropogenic aerosols have changed considerably in
these regions during the 20th century (e.g. Lamarque et al., 2010).
The emission regions (North America - NA, Europe - EU, South Asia - SA and East Asia - EA) are defined according
to the updated region definition of the Task Force on Hemispheric Transport of Air Pollution (HTAP), see Fig. 1, and the
aerosol emissions are the historical emissions of CMIP5 described by Lamarque et al. (2010). The aerosol type we study here
is ammonium sulphates and thus we perturb the anthropogenic sulphur dioxide ($SO_2$) emissions provided for CMIP5.
Year 2000 is chosen as the baseline year and aerosol emissions, aerosol precursor emissions, trace gas concentrations and
land use representing this year are used for the control simulation. In the emission perturbation experiments, the anthropogenic
aerosol precursor emissions are decreased or increased compared to year 2000 emissions and kept constant in each region
throughout the simulation. In total five coupled sensitivity experiments were performed, four experiments where $SO_2$ emissions
were increased in the four different regions and one where anthropogenic $SO_2$ emissions were removed over Europe. The
simulations were started from year 2000 in the transient historical CMIP5 simulation. The simulation length is 160 years for
simulations where emissions are increased. For the experiment where emissions are decreased the simulation length is 200
years. All the results presented are annual mean quantities and the first 50 years of each simulations have been removed before
averaging.
The $SO_2$ emission changes in the emission perturbation experiments are shown in Fig. 2. In the 0xEU experiment the $SO_2$
emissions in Europe are not completely eliminated. There remains 4.66 $Tgyr^{-1}$ of volcanic emissions of $SO_2$ in Europe (from
Etna). The $SO_2$ emissions in the rest of the experiments were increased by varying amounts depending on the magnitude of
the regional emissions in the control simulation. This was done to obtain a global mean instantaneous radiative forcing of
approximately -0.45 $Wm^{-2}$ in all these perturbation experiments. For South Asian emissions, which are low in the control
simulation (6.47 $Tgyr^{-1}$ in year 2000 compared with 24.53 $Tgyr^{-1}$ in East Asia) the emissions were increased by a factor
of ten. Similarly, for Europe, North America and East Asia, $SO_2$ emissions were increased by a factor of seven, five and five
respectively.
The 0xEU experiment is included so that the effect of emission perturbation magnitude can be investigated, i.e. the sensitivity
to a relatively small emission reduction compared to a relatively large emission increase. The emission perturbation magnitude
(and sign, i.e. reduction) could also be considered as a more likely future scenario.



With the resulting global temperature response field of each emission perturbation experiment, RTP coefficients, $dT_r/dEm_e$, can be constructed relating emission changes in the predefined emission regions, $e$, to any response region, $r$, of choice. The emission-based ARTP can be calculated from the absolute emission change:

$$ARTP_{r,e}^{EM} = \Delta Em_e \frac{dT_r}{dEm_e}. \tag{3}$$

In addition to the coupled experiments we perform simulations to evaluate the Instantaneous Radiative Forcing (RF) and Effective Radiative Forcing (ERF) of the aerosol emission perturbations in the coupled experiments.

The RF is derived from fixed Sea Surface Temperatures (SSTs) simulations where dual calls are made to the radiation code: one call with climatological aerosols and another call where the emission perturbation aerosol concentrations and their effect on cloud albedo are sent to the radiation code solely for diagnosing the radiative effect of these. Thus the meteorology in the RF simulations is identical since the radiative effects of the emission perturbations do not feedback on the meteorology. With this methodology the radiative effects alone from the aerosol can be quantified, without influence of fast or slow feedbacks. The RF simulations are 7 years long and the 5 last years are used for the analysis.

The ERF is derived by performing fixed SST simulations with aerosol emission perturbations and letting the radiation changes affect the meteorology. Thus, in addition to the aerosol direct radiative effect and cloud albedo effect the ERF also includes radiative changes from fast feedbacks such as cloud microphysical and semidirect effects. In NorESM these effects includes e.g. cloud liquid water content cloud fraction. These simulations are 20 years and the 15 last years are used for the analysis.

In a simplified manner, the process chain from emission to global mean temperature response can be thought of a translation of emission to column burden, to the instantaneous direct and indirect radiative forcing, to forcing including fast feedbacks, to the full coupled temperature response. In an attempt to identify where the largest divergence appears in the process chain from emission to temperature response in the experiments conducted with NorESM, we investigate the usefulness and accuracy of alternative quantities to the unit emission in predicting the surface temperature response.

## 3 Results

### 3.1 Global forcing and temperature response

The simplest way to describe the sulphur emission perturbation impact on global and regional temperatures is to express the temperature response in terms of temperature change per unit emission of sulphur (cf. Sect. 2.1). We first analyse the results from the sensitivity experiments where $SO_2$ emissions were increased. The results from the 0xEU experiment will be discussed in Sect. 3.3. The global mean temperature response per unit emission for these sensitivity experiments where the $SO_2$ emissions were increased by comparable magnitudes the global temperature change per unit emission is similar within 10%. The temperature response varies from -0.0056 to -0.0061 $K(TgSyr^{-1})^{-1}$, depending on the location and magnitude of the sulphur emission change (Table 2).


All global mean temperature changes are significantly different compared to the temperature of the year 2000 control simu-
lation, but are not significantly different between each other (Fig. 3a). Thus, the location of an emission change does not appear
to be a governing factor for the global mean temperature response modelled by NorESM. However, all emission changes are
located in the northern hemisphere, and atmospheric transport of aerosol particles will contribute to a redistribution of atmo-
spheric concentrations and the resulting column burden and radiative forcing of the aerosol, so that in some cases the resulting
column burden and radiative forcing from emission changes in different regions will partly overlap.
The global average RF per unit emission change (Table 2) shows larger variability than the global temperature response
(varying from -0.010 to -0.017 $\mathrm{Wm}^{-2}(\mathrm{TgSyr}^{-1})^{-1}$, the largest RF value being 62% larger than the smallest value), a larger
emission change is needed in EU than in SA to obtain the same RF change. The variability for the global mean ERF is similar
to that of the RF (difference of 64% between the largest and smallest value, varying from -0.008 to -0.026 $\mathrm{Wm}^{-2}(\mathrm{TgSyr}^{-1})^{-1}$)
but the magnitude of the global mean ERF is smaller than the RF for all emission-increase experiments except for the 5xNA
experiment. Thus, on a global scale, fast cloud feedbacks contribute to dampen the forcing effect of the emission increases in
NorESM.

### 3.1.1   Emission changes as predictor of global mean temperature change

As outlined in Sect. 2, the extreme simplification inherent in the method of describing the temperature response in terms of
emission perturbations, leads to uncertainties related to the generality of the RTP coefficients.
Figure 4a illustrates how $SO_2$ emission perturbations in the different experiments translate to global sulphate column burden,
RF, ERF and temperature anomalies. All values are normalised by the response in the North American experiments to illustrate
the relative amount of variability for each response quantity (i.e. response in the 5xNA experiment is always one in Figure 4.)
As noted previously, the global temperature responses per unit emission in the experiments where $SO_2$ emissions are in-
creased are not significantly different from each other. However, the translation from emission to column burden shows a
different pattern. For this quantity, the column burden per unit emission in the 10xSA experiment is 76% higher than in the
other experiments. Thus, the geographical location seems to be one factor controlling the column burden sensitivity to emis-
sion perturbations in the experiments where emissions are increased. The increased emissions in SA together with a local SA
reduction in precipitation of 0.22 $\mathrm{mmday}^{-1}$ lead to a longer residence time of sulphate (0.73 days longer) as well as other
aerosol particles in NorESM in the 10xSA experiment compared to the control experiment.
A similar pattern as the column burden is evident for the normalised instantaneous RF response to a unit emission change.
The RF response to a unit emission change in SA is larger than the responses in the other experiments. Thus, there appears
to be a close connection between changes in the global sulphate column burden and the RF (correlation coefficient $r$=-0.985).
The normalised ERF sensitivity to unit emission perturbations, shows a larger variability between the experiments compared to
the other investigated quantities. The standard deviations for the global average ERF responses are also larger than that for RF.
This result indicates that cloud feedbacks, such as changes in liquid water content or cloud fraction and cloud albedo contribute
substantially to the ERF (cf. Table S1 in supplementary material), and also contributes to larger variability.



Figure 4b shows the temperature response normalised by the different "basis quantities" (i.e. the leftmost group of bars in Fig. 4b are identical to the rightmost bars in Fig. 4a). The perfect basis quantity would be one for which the heights of all bars corresponding to the different experiments are equal. A basis quantity with this property would be the ideal predictor of the global mean temperature response. Figure 4b shows that emission perturbation is a good predictor of the temperature response for emission increases from all regions investigated when emissions are increased in all regions (standard deviations corresponding the each group of bars are presented in Table 3). Instantaneous RF and column burden as basis quantities underestimate the temperature response to SA emissions (this is connected to the larger column burden and RF sensitivity to a unit emission perturbation in SA which do not translate to a larger temperature sensitivity). For ERF there is substantial variability in the predictability for the temperature responses in the emission increase experiments, which also yields the largest standard deviation of the basis quantities for these experiments. Thus, emission is a better predictor than the ERF of the global temperature response for these emission increase experiments.

## 3.2 Sub-global forcing and temperature response

### 3.2.1 Latitudinal forcing and temperature response

The sub-global normalised temperature responses in the experiments where $SO_2$ emissions were increased display more variation between the different experiments than the global mean sensitivities. (As mentioned before, the 0xEU experiment will be discussed in Sect. 3.3.2.) The latitudinal temperature responses per unit emission in the experiments with increased emissions show a qualitatively similar pattern of increasing sensitivity with increasing latitude (Fig. 5). This pattern of Arctic amplification is not dependent on the location of the emission perturbation in these experiments, neither in the latitudinal nor the longitudinal direction. The temperature responses in each latitude band are significantly different from the temperature in the year 2000 control simulation (at the 99% confidence level), except for the southern hemisphere temperature responses (indicated by gray shading of the columns in Fig. 5). The latitudinal temperature responses in the different experiments are not significantly different from each other, with the exception of most of the latitudinal temperature responses to SA emissions (at the 90% confidence level, see Fig. 3 for details). Thus, the latitudinal temperature responses are in principle indistinguishable for emission increases from EU, NA and EA, while the SA emission response is generally weaker.

The only latitudinal RF and ERF that are statistically significant are the responses to emissions increases in EU, NA and EA, in NHml, the latitudinal band inside which these emission regions are located. Significant ERF responses are also found in ARCT for the same emission source regions, but the ERF is larger in NHml where the emissions changes are located, than in ARCT. $SO_2$ emissions increases in SA do not lead to any latitudinal average RF or ERF response that are statistically significant. A large fraction of the atmospheric sulphur mass from SA emissions (which are mainly emitted in the Tropics) is transported to the NHml region, so that the average RF, ERF and column burden in this region exceeds that of the tropical region. However, the total integrated sulphur column burden is larger in the Tropics than in the NHml (not shown) in the 10xSA experiment.





The ERF acts to enhance the forcing relative to the RF in the NHml in all experiments, as well as in the ARCT region.
This is a manifestation of aerosol indirect effects which lead to e.g. higher cloud water content (Table S1). The ERF displays a
warming effect in the SHext in all experiments (due to decreases in low cloud fraction at southern hemisphere midlatitudes, not
shown), although this positive ERF is not significant in any experiment. However, the positive ERF in the southern hemisphere,
which represents a large part of the global mean, contributes to the lower value of global average of the ERF compared to the
RF (cf. Sect. 3.1).
As described above, the temperature responses in the latitudinal bands are similar between the experiments with the excep-
tion of the temperature responses to changed $SO_2$ emissions in SA. SA has the largest tropical response which, however, is only
significantly different from the tropical response to EU emissions, which is the weakest tropical response among the experi-
ments. Similarly, the ARCT response to SA emissions is the smallest among the experiments, and is only significantly different
to the ARCT response to NA emissions, which leads to the strongest response in ARCT. The weaker NHml response to SA
emissions compared to the other emission regions, on the other hand, is significantly different compared to all other NHml
temperature responses. The NA, EU and EA emission regions are to the greater part located in the northern hemisphere mid-
latitudes, and mostly north of the SA emission region. Thus, the longitudinal position of a mid-latitude emission perturbation
does not appear to matter for the latitude mean temperature responses at northern hemisphere high- and mid-latitudes.
### 3.2.2   Regional temperature response
The differences between the sub-global temperature responses in the different experiments become more evident when they
are derived for the emission perturbation regions (and the AR region north of 66°N) compared to when derived for latitudinal
bands (Fig. 7). All regional temperature changes are statistically significant compared to the control simulation. The largest
temperature response is found in the AR region in all experiments, which is consistent with the latitudinal distribution of
the temperature response for latitude bands described in the previous section. Similarly, the SA emissions have the smallest
effect on the AR temperature among the experiments, but the AR temperature response in this experiment is only significantly
different from the response to NA emissions, which give the largest AR response among the experiments.
Outside the AR region, the largest temperature response is found locally in the emission region in all experiments except
10xSA. This result is consistent with the forcing always being largest in the emission region (Fig. 8). The regional RF and ERF
is also statistically significant for local $SO_2$ emissions from SA, as opposed to when derived for the Tropical latitudinal band
(Fig. 6). For SA emissions the temperature response in the EA region is marginally larger than the local temperature response
in the SA region. The EA region is located downwind of the SA region, which means that a substantial part of the sulphur
emitted in SA is transported to EA and contribute to the local forcing in EA. The column burden increase by $3\%/TgSyr^{-1}$ in
EA due to SA emission, to be compared with the increase in EA due to local emission of $4\%/TgSyr^{-1}$. Additionally, advection
of air originating from SA might also partly explain the large temperature response in the EA region to SA emissions. EA is the
only region where there are emissions from a remote region (SA) that lead to a temperature response that is indistinguishable
from the effect of local emissions.



The local temperature responses in the emission perturbation regions are larger than the corresponding zonal mean temper-
ature responses of the latitudes covered by each region (indicated by black dots in Fig. 7) in all experiments. The largest local
response relative to the zonal mean is found in the 10xSA experiment, which is 66% larger than the zonal mean. The 5xNA
experiment shows the largest absolute difference between the local response and the zonal mean, 0.0055 K/TgSyr$^{-1}$ (55%
larger). The smallest local temperature response relative the zonal mean is found for 7xEU (20%). All differences between
these local responses and the corresponding zonal means are statistically significant at the 95% confidence level.
For both NA and EU emission perturbations, the temperature responses in the regions outside the emission regions are close
to the corresponding zonal mean responses (within 2-17% difference). SA and EA emission perturbations, on the other hand,
both lead to a larger temperature response than the corresponding zonal mean for NA and a smaller temperature response than
the zonal mean for EU, where both these differences between the zonal mean and regional temperature response are statistically
significant. Both EA and SA emission perturbations have a substantial effect on NA temperature, of the same magnitude as the
local responses for these emission regions, despite the geographical distance between the emission location and the temperature
response regions. Local radiative forcing in NA is not responsible for this temperature effect (Fig. 8). This result points towards
a far field effect in the temperature response to Asian aerosol forcing which is mediated by atmospheric circulation changes
rather than radiation changes.

## 3.3   Nonlinearities

So far, only the results from the experiments where $SO_2$ emissions were increased have been discussed. In this section we will
focus on the differences between the results from the 0xEU and 7xEU $SO_2$ emission changes experiments. The purpose is to
investigate if the emission perturbation magnitude or background state influences the temperature response (cf. e.g. Wilcox
et al., 2015).

### 3.3.1   Global temperature response

In the experiment where European anthropogenic $SO_2$ emissions are removed, the global average temperature change per unit
emission is approximately twice of that in the 7xEU experiment, as well as in the other experiments where emissions were
increased (Fig. 4 and Table 2). This indicates that there is a non-linearity depending on the magnitude and sign of the emission
change, at least for European $SO_2$ emissions. Since the coupled simulations include aerosol indirect effects, and since indirect
effects are usually larger than direct aerosol effects (Rap et al., 2013; Myhre et al., 2013; Kirkevåg et al., 2013), nonlinear
effects pertaining to aerosol-cloud interactions most likely play a role in the difference in global climate sensitivity between
the 0xEU and 7xEU experiments. However, effects related to the modeled aerosol microphysics could also play a role in this
difference, in particular when $SO_2$ emissions and concentrations are low. E.g. in extreme conditions the partitioning between
different aerosol microphysical paths might change, like condensation and nucleation rates of sulphate (Stier et al., 2006).
The two experiments with different European $SO_2$ emission perturbations illustrate the difficulties related to the generality
of the method of translating emission perturbations to temperature response already discussed in Section 3.1. The global mean





temperature responses per unit sulphur emission differ substantially for these two experiments, as well as the magnitudes of
the latitudinal and regional temperature responses.
We return to the question of "basis quantities" (cf. Sect. 3.1) and for which step in the translation from emission to tem-
perature response the largest divergence appears for the different experiments. The normalised global temperature responses
per unit emission in the experiments where $SO_2$ emissions are increased are close to unity, while the normalised temperature
response per unit emission in the 0xEU experiment is larger than two (Fig. 4). The translation from emission to column burden
for the EU emission changes is not dependent on the emission magnitude in the experiments presented here. Similar to what
was noted for the other experiments, the RF per unit emission change in 0xEU and 7xEU is similar to the column burden
response per unit emission change. The normalised ERF sensitivity to unit emission perturbation on the other hand, bears more
resemblance with the temperature response for the 0xEU and 7xEU experiments (third group of bars/the next rightmost bars
in Fig. 4a). This indicates that fast cloud feedbacks, such as cloud lifetime, liquid water content or semidirect effects, is most
likely a key component for understanding the non-linearity in the temperature response to European emissions (cf. Table S1
and S2).
Emission perturbation was in Sect. 3.1 found to be a a good predictor of the temperature response for emission increases
from all regions investigated when the emissions were increased with similar magnitudes. However, it does not capture the non-
linear behaviour in the temperature response to European emission perturbations of different magnitudes (Fig. 4b). Similarly,
RF and column burden as basis quantities also fail to capture this property in the response to European emission perturbations.
The ERF is the only basis quantity that captures the non-linearity for European emission perturbations of varying magnitude.
However, there is substantial variability in the predictability for the temperature responses in the other experiments. The ERF
shows the smallest standard deviation for the different basis quantities when all experiments are considered (Table 3), but this
is due to substantially larger standard deviations for emissions, CB and RF as basis quantities when the 0xEU experiment is
included. Nevertheless, the ERF is the basis quantity with the highest degree of generality for the global results from all the
experiments conducted with NorESM presented in this study.
**3.3.2 Sub-global temperature response**
Similarly to the global mean response, the magnitude of the latitudinal and regional temperature responses per unit sulphur
emission are substantially larger in the 0xEU experiment than in the 7xEU experiment, with the exception for the temperature
difference in SA which is not statistically significant compared to the control simulation (Fig. 9, where that hatched bars
indicate the 7xEU response for easy comparison). For the latitudinal sensitivities, the pattern of increasing temperature response
with latitude found in the experiments where emissions were increased (Sect. 3.2.1 and Fig. 5) is also seen for the 0xEU
experiment. The relative impact on the southern hemisphere is also larger in this experiment compared to the other experiments.
All latitudinal temperature changes in the 0xEU experiment are significantly different from the responses in all the other
experiments except for the tropical latitude band (Fig. 3).
The regional 0xEU responses display a similar pattern to the regional responses in the 7xEU experiment, but with different
magnitudes. The largest temperature response is seen in the AR region whereas outside AR the largest response is found in the





emission region (EU). The temperature responses to reduced EU SO$_2$ emissions in NA and EA are close to the zonal means
for the latitudes covered by these regions (within 2%). This is similar to the the corresponding regional temperature responses
in the 7xEU experiment relative to the zonal mean responses.
The non-linear effects are mostly confined to the magnitude of the temperature responses in the case for European emission
perturbations in these experiments. Zonal asymmetries do not appear to have a significant impact on the regional temperature
responses. This might, however, be different for the Asian emission perturbations where zonal asymmetries seem to play a more
prominent role in the regional temperature distributions compared to the European and North American emission perturbations.

### 3.4   Comparison with other RTP coefficients

In this work we have aimed to establish the simplest possible model for anthropogenic aerosol impacts on regional tempera-
tures, i.e. an emission-based regional temperature potential coefficient.
Nevertheless, difficulties associated with nonlinear effects in this relationship remain where ERF proved to be a more general
basis quantity for estimating the global temperature response than emissions, in terms of capturing different magnitudes of
global mean temperature responses for different emission changes in Europe.
With the experimental set up applied in this study, it is not possible to derive sub-global (latitudinal or regional) radia-
tive forcing-based sensitivities, as the forcing changes in the different experiments are not confined to a certain region or
latitude band. However, with the latitudinal and regional RF and ERF from the different experiments, the generality of the
RTP-coefficients derived by Shindell and Faluvegi (2009) and Shindell (2012) can be assessed for the NorESM generated
temperature response. For each experiment the RF and ERF in each latitude band resulting from the regional emission per-
turbations are calculated (Table 4) and used with different methods for calculating the latitudinal temperature responses, the
ARTP.
First we compare the temperature response as calculated from Equations 1 and 2 with that from the simulations with NorESM
where SO$_2$ emissions were increased. Both equations require knowledge of the model global climate sensitivity (or the Impulse
Response Function). The climate sensitivities are derived from the emission perturbation experiments, and we use a mean value
from all experiments with emission increases. Climate sensitivities for both RF and ERF are derived, and these are calculated
to be 0.47 and 0.61 K(Wm$^{-2}$)$^{-1}$, respectively.
However, the model global climate sensitivity is not always known, e.g. if the forcing is derived with a Chemistry Trans-
port Model (CTM). Moreover, one motivation behind using RTP coefficients is to avoid conducting multi-century coupled
simulation, which is necessary for deriving the climate sensitivity. Therefore, we also evaluate the performance of the RTP
coefficients with a standardised climate sensitivity as well as applying the RTP coefficients of Shindell and Faluvegi (2010)
as regional sensitivity coefficients (i.e. without normalising with the regional climate sensitivity to global forcing and scaling
with the models global sensitivity). This is to see how well the RTP-method predicts the model temperature response when the
specific model's climate sensitivity to a particular forcing agent is unknown.
The latitudinal temperature responses calculated from Equations 1 and 2 are shown in Fig. 10 and 11. The small dots indicate
the temperature response in specific regions and the filled circles indicate the emission source regions. The high latitude



temperature response in the northern hemisphere (ARCT) calculated using the RTP coefficients, the ARTP, is underestimated
compared to the temperature response in the NorESM experiments (but still within one standard deviation of the NorESM
simulated temperature response), except for when the ERF is used in combination with the normalised coefficients of Shindell
et al. (2012) (Fig. 11b). This is also the method that gives the smallest root mean square deviation (RMSD) of 0.14K (RMSDs
are displayed in each panel). In general, ERF is a better predictor of the latitudinal temperature response than RF, based on
the RMSD. Similarly, the RTP coefficients that are normalised by the global sensitivity (Shindell et al., 2012) rather that the
regional sensitivity (Shindell and Faluvegi, 2010), i.e. Fig. 10 vs. Fig. 11, is a better model for the temperature response in each
latitude band, also based on the RMSD. This was also pointed out by Shindell (2012).
However, the performance of this method relies on that the correct climate sensitivity is used and is known. The standard
definition of equilibrium climate sensitivity (ECS) is the equilibrium temperature response to a doubling of $CO_2$ (Collins et al.,
2013), and is available for nearly all models participating in the Coupled Model Intercomparison Project phase 5 (Flato et al.,
2013). For NorESM this climate sensitivity has been estimated to 1.01 $Wm^{-2}K^{-1}$ (Iversen et al., 2013). This is higher than
the sensitivity to aerosol forcing obtained in this study. The results of the RTP method with this ECS applied is shown in Figure
12. Overall, the use of ECS overestimates the temperature response in almost all latitude bands. Thus, it is important to use the
correct climate sensitivity for the climate forcer investigated. This is a complicating factor since it requires a priori knowledge
of this quantity, which can only be derived by performing coupled simulations, the necessity of which one often would like
to eliminate with a simplified method. Moreover, if calculations to derive radiative forcing are performed with a CTM, this
quantity is not available.
A third alternative is to apply the RTP coefficients without normalising with a model dependent climate sensitivity parameter
(Fig. 13). The implicit assumption in this method is that the sensitivity of NorESM to aerosol forcing is equal to that of the
GISS model in Equation 2, a transient sensitivity of 0.5 $K/Wm^{-2}$ (Shindell, 2012). This assumption about the sensitivity leads
to RTP-derived temperature responses with smaller RMSD values than both those derived by applying the ECS for NorESM
in Equation 1 and 2.
Figure 13 shows that assuming that the sulphate aerosol climate sensitivity is similar between different climate models might
be better than assuming that the climate sensitivity for sulphate aerosol is similar to the ECS derived from the same model.

## 4   Discussion

### 4.1   Uncertainties associated with RTP coefficients

The method applied in this work, i.e. evaluating the global and regional temperature responses based on the emission change
magnitudes, means that on the one hand, the starting quantity is easy to assess and compare and is easy to incorporate into
integrated assessment models, such as GAINS. The full response chain from emissions to atmospheric concentrations, to forc-
ing, to surface temperature response is accounted for in this metric. On the other hand, the fact that the metric encompasses
the full chain from emission to temperature response means that there are implicit uncertainties in the metric. The representa-
tiveness of these emission based RTP coefficients will depend on how well the climate model used to derive these coefficients,





represents a large number of atmospheric chemical and physical processes on many different spatial and temporal scales. The
RTP coefficients derived by Shindell and Faluvegi (2009) and Shindell (2012) were derived from radiative forcing, and thus
do not contain the uncertainties introduced when estimating the column burden and forcing associated with aerosol emissions.
However, a model to translate emission to radiative forcing, either RF or ERF, is still necessary to make these forcing based
RTP coefficients useful in an integrated assessment modelling context based on emission pathways.

Some major uncertainties can be identified if the emission-temperature response chain is broken down into sub steps. First,

emissions of an atmospheric chemical compound result in an atmospheric concentration and column burden. The translation
from emission of an atmospheric chemical component to atmospheric aerosol loading depends on a number of factors, e.g. if the
aerosol originates from primary emission or is formed through chemical reactions in the atmosphere (i.e. secondary aerosols),
like sulphate which is studied here. The aerosol production for secondary aerosols will depend on which and how chemical
reactions that produce these aerosols are described in the atmospheric model. Kasoar et al. (2016) found that the efficiency of
chemical conversion of $SO_2$ to sulphate was one process contributing to differences in the simulated responses in three different
climate models to equivalent emission reductions over China. In addition to chemical production, the interaction with clouds
will influence the atmospheric concentration of aerosols. Wet removal through precipitation is an efficient removal process
for hygroscopic aerosols like sulphate containing compounds. All these factors, emission strength, atmospheric production
and removal efficiency influence how long aerosol particles stay in the atmosphere and how far they are transported from the
emission sources. Thus, all these processes influence the atmospheric loading and how these processes are represented in the
model will influence the modelled aerosol column burden.

Another source of uncertainty in the emission-forcing-temperature chain, besides the modelled column burden, is how the

aerosol radiative properties are modelled (Myhre et al., 2013). The radiative properties of aerosols depend on e.g. their chemical
composition, water content and mixing state. Thus, given the same atmospheric concentration and distribution of aerosols, their
radiative effect might vary depending on how their radiative properties are represented in the model. Other complicating factors
when it comes to aerosol radiative effects are clouds and aerosol indirect and semi-direct effects on clouds. The direct radiative
forcing will depend on the cloud distribution itself, and aerosol can affect the properties of clouds and also, affect the cloud
distribution, i.e. other components, besides the aerosol itself, within the model influence their radiative effects (Stier et al.,

2013).

One of the largest uncertainties associated with the effect of aerosols on climate is related to their indirect effect on clouds

(Myhre et al., 2013) and the representation of these can vary widely between different models. Beside chemical conversion
and radiative impacts, Kasoar et al. (2016) also identified indirect effects on clouds as a major source of diversity between the
models they investigated. Wilcox et al. (2015) found that parameterisations of the relationship between cloud droplet number
concentration and effective radius was the largest contribution to differences in the cloud albedo effect between three models
from the CMIP5 archive, among those NorESM.

The factors described above all contribute to inter-model diversity, and will influence how general RTP coefficients are

across models. However, the same processes also contribute to regional sensitivity differences within the same model, but not





based on differences in how the processes are represented in the model, but on the specific meteorological conditions in each
region (e.g. cloud climatology, regional circulation patterns and the background aerosol).
It is evident from the results presented in this study that the temperature sensitivity depends on the emission change magni-
tude in NorESM. The nonlinearity in the response appears to belong to aerosol interactions with clouds and in particular to fast
feedbacks included in the ERF. These include changes in liquid water content, cloud fraction and subsequent changes in cloud
albedo of the new cloud distribution, i.e. cloud life time effects (Albrecht, 1989) (the cloud albedo effect of the background
cloud distribution is included in RF).
Wilcox et al. (2015) derived simple functional forms representing the relationship between sulphate load and cloud droplet
effective radius (i.e. the cloud albedo effect) in three different CMIP5 models, with which they could reproduce the time
evolution of the simulated cloud droplet effective radius from historical 20th century simulations. With these functional forms,
they could also quantify the intrinsic varying sensitivity in the parameterisation of the effective radius which depends on the
magnitude of the sulphate load, and how the effective radius (and ultimately radiative forcing) goes from being highly sensitive
at low sulphate loads to a relative insensitive state at high sulphate loads. While they focussed on the cloud albedo effect, the
cloud life time effect is a direct consequence of initial change in effective radius, and should thus display a similar varying
sensitivity depending on the absolute sulphate load.
Thus, the similarity of the global temperature responses in the emission increase experiments, despite different mechanisms,
might be due to this saturation of cloud droplet effective radius change when emission increases are large enough. The tem-
perature sensitivity for the different regions could prove to be different if emission were reduced, even by equivalent amounts,
depending on the regional background emission strength and regional meteorological conditions. Nonlinear effects depending
on the emission change magnitude and background is one of the biggest hurdles in creating a general emission based RTP
coefficient.
**4.2  Basis quantity**
Different quantities for predicting the temperature response have been assessed for the global mean temperature and for latitu-
dinal bands in combination with the RTP coefficients of Shindell and Faluvegi (2010) and Shindell (2012). In both cases ERF
proved to have the best skills to predict the temperature response.
For the global mean temperature response, the ERF was the only variable that was capable of capturing the large difference
in the temperature responses to the European increase and decrease in $SO_2$ emissions. However, for the emission increase
experiments, emission was the quantity that best predicted the temperature change. Also for the latitudinal ARTPs the ERF
performed better in predicting temperature responses than the RF for NorESM, which is mostly due to a simulated larger ERF
than RF in the Arctic region. This can either be an indication that the sensitivity of the Arctic region is larger in NorESM than
GISS to forcing outside the Arctic region, i.e. that the coefficient relating the forcing to Arctic temperature responses should
be larger for NorESM. It could also be an indication that the cloud feedbacks in the Arctic is a necessary part of the forcing,
and that the local forcing from fast feedbacks is important for the Arctic response in NorESM.





## 4.3 Latitudinal and regional sensitivities

The sensitivity of zonal mean temperatures to emission perturbations in different regions show large similarities, with the exception of the overall weaker northern hemisphere temperature response to SA $SO_2$ emissions; the zonal mean temperature change increase with increasing latitude in all experiments and do not appear to depend strongly on the location of the emission perturbations within the northern hemisphere (Fig. 5). There are many factors that might contribute to the weaker temperature response to the SA emission perturbation. This emission perturbation is located in one of the major monsoonal regions on the globe, and the increase of sulphate leads to a substantial reduction of precipitation over SA (Table S1 and S2). The reduced precipitation, in turn, leads to less efficient wet removal of aerosol resulting in an increased residence time and a larger column burden response per unit emission of both sulphate and BC compared to the control simulation. The decrease in precipitation in SA (as well as smaller increases in liquid water path, Table S1 and S2) also contribute to a weaker ERF and indirect effect on clouds, which, in the other experiments enhances the local forcing, but not in SA (Fig. 8). This result is one example of how different local meteorological conditions where the emission changes occur contribute to different forcing and temperature responses within the same model.

The general pattern, which indicates a stronger temperature response with increasing latitude for all emission perturbations, is a robust feature in all experiments. In all experiments, the second largest regional sensitivity (after the Arctic region), is generally found in the region of the emission perturbation. However, for SA emissions, the sensitivity is slightly larger in the East Asian region compared to the South Asian emission region, a result caused by production of sulphate aerosol from $SO_2$ and subsequent transport from SA to EA.

Moreover, Asian $SO_2$ emissions, both from EA and SA, produce larger zonal asymmetries in the global temperature change field than those of EU and NA. The Asian $SO_2$ emissions lead to temperature responses in NA and EU that are higher and lower, respectively, than the zonal mean response. The remote regional temperature responses to EU and NA $SO_2$ emissions are on the other hand close to the corresponding zonal mean responses. The location in the Asian monsoon region and proximity to the Western Pacific mean that these $SO_2$ emissions could cause tropical precipitation changes that are effective in generating planetary scale waves. These waves can propagate into the extratropics, which in turn influences the global temperature distribution (Ming et al., 2011; Lewinschal et al., 2013).

However, the standard deviations for the regional sensitivities are larger than those for the latitudinal sensitivities and zonal mean sensitivities. Nevertheless, despite the larger uncertainties associated with the regional RTPs compared to the latitudinal RTPs, they provide information that is not captured by the latitudinal RTPs.

## 5 Summary and Conclusions

We performed simulations with the Earth system model NorESM to evaluate the surface temperature change in response to $SO_2$ emission perturbations in Europe, North America and East and South Asia, and to derive emission-based RTP coefficients. Four experiments were performed where emissions were increased relative to the year 2000 in each individual region to yield





similar global mean radiative forcing values. One additional experiment was performed where anthropogenic $SO_2$ emissions
were completely removed in Europe.
In all five experiments the zonal mean latitudinal temperature change distribution showed a similar pattern of increasing
temperature change with increasing latitude, independently of where the emission perturbation was located. The largest tem-
perature response in all experiments performed was in this study thus found in the Arctic region, no matter where the emission
perturbations were located. Outside the Arctic region, the temperature response was largest in the emission perturbation re-
gion, except for SA emissions where the temperature response in the neighbouring EA region was equally large. This result
was consistent with the radiative forcing pattern, which was also strongest in the emission region in each experiment.
Furthermore, it was found that the emission-based RTPs derived with NorESM are non-linear. Removal of anthropogenic
European $SO_2$ emissions led to a temperature response per unit emission approximately twice of that in the 7xEU experiment.
Other differences were also noticed for the regional responses to regional emission perturbations. Asian emission increases led
to a different remote effect compared to increases in EU and NA emissions. Both EA and SA emission perturbations led to
a NA temperature response that was larger than the zonal mean and an EU response that was smaller than the corresponding
zonal mean. EU and NA emission perturbations, on the other hand, led to remote responses that were close to the zonal mean
for the same latitudes.
A comparison of the modelled temperature response in NorESM with that calculated using ARTPs (equations 1 and 2)
derived with the RTP coefficients of Shindell and Faluvegi (2010) and Shindell (2012) showed that the RTP coefficients predict
similar latitudinal temperature change distributions as those produced by NorESM. The agreement between the calculated
values using ARTPs and the temperature change simulated using NorESM was better when ERF was used together with the
RTP coefficient than when RF was used. This was mainly due to a larger Arctic ERF than RF that resulted in an Arctic
temperature response closer to that produced in the NorESM simulations. This result could be an indication that the Arctic is
more sensitive to forcing outside this region in NorESM than in the GISS model, or that local fast cloud feedbacks are crucial
for the Arctic temperature response in NorESM.
Even though the global mean temperature response to emission increases is similar in all regions, the processes leading to
the change may be different in different regions, as it depends on the local meteorological conditions. In all regions except SA,
aerosol indirect effects on clouds, and particularly life time effects, are dominating the ERF response. For SA, direct radiative
effects have a higher relative importance in the response since the local responses in cloud fraction, liquid water path and
precipitation are either weaker compared to the other emission regions or decrease in response to increased $SO_2$ emissions.
The latitudinal distribution of the zonal mean temperature response to SA emission changes also differs from the rest of the
simulations in that the Northern hemisphere response is weaker and the southern hemisphere and tropical responses are stronger
than in the other simulations.
Air pollution globally cause more than 4 million premature deaths each year and as sulphates are major air pollution com-
ponents, emission reductions of $SO_2$ will be absolutely necessary to improve air quality. The derived emission-based RTPs
will simplify development of cost effective co-beneficial abatement strategies that can give both better air quality and mitigate
climate change. The nonlinear effect predicted by NorESM indicate a reduced immediate climate effect of $SO_2$ emission re-




ductions in highly polluted areas where the indirect effect is saturated but the effect would become more evident with time as
the saturation of aerosol indirect effects diminishes. Nevertheless, emission reductions of $SO_2$ and other short-lived climate
forcers are necessary for improving air quality and public health in both Europe, North America and Asia.
*Author contributions.* AL, AMLE, HCH, MS, TKB and JL designed the experiments. AL carried out the simulations. AL prepared the
manuscript with contributions from all co-authors.
*Competing interests.* The authors declare that they have no conflict of interest.
*Acknowledgements.* This work was supported by the Swedish Environmental Protection Agency through the Swedish Clean Air and Climate
research program (SCAC). The NorESM simulations were performed on resources provided by the Swedish National Infrastructure for
Computing (SNIC) at the National Supercomputer Centre (NSC).



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



**Table 1.** Latitudinal bands definition and region definitions.

| Name | Latitudes or region definition |
|------|-------------------------------|
| SHext | 90°S-28°S |
| Tropics | 28°S-28°N |
| NHml | 28°N-60°N |
| ARCT | 60°N-90°N |
| AR | 66°N-90°N |
| EU | Europe - HTAPv2 |
| NA | North America - HTAPv2 |
| EA | East Asia - HTAPv2 |
| SA | South Asia - HTAPv2 |



**Table 2.** Global results from the experiment where SO$_2$ emissions in different regions are changed. Units are $10^{-2}$K/TgSyr$^{-1}$ for temperature and $10^{-2}$Wm$^{-2}$/TgSyr$^{-1}$ for RF and ERF.

| Experiment | 0xEU SO$_2$ | 7xEU SO$_2$ | 5xNA SO$_2$ | 5xEA SO$_2$ | 10xSA SO$_2$ |
|---|---|---|---|---|---|
| $\Delta$T/$\Delta$em | -1.28 | -0.56 | -0.61 | -0.58 | -0.58 |
| RF/$\Delta$em | -1.30 | -1.04 | -1.22 | -1.14 | -1.68 |
| ERF/$\Delta$em | -2.55 | -0.78 | -1.29 | -1.00 | -0.88 |



**Table 3.** Standard deviations for the different normalised basis quantities evaluated in Figure 4b (unitless).

| Variable | EM | IRF | ERF | CB |
|---|---|---|---|---|
| Increased emissions | 0.03 | 0.15 | 0.19 | 0.17 |
| All experiments | 0.46 | 0.43 | 0.19 | 0.51 |



**Table 4.** Regional radiative forcing (RF) and effective radiative forcing (ERF) in $Wm^{-2}$ used to derive latitudinal ARTPs in Fig. 10-13.

| Experiment | 0xEU $SO_2$ | 7xEU $SO_2$ | 5xNA $SO_2$ | 5xEA $SO_2$ | 10xSA $SO_2$ |
|---|---|---|---|---|---|
| RF | | | | | |
| SH | 0.000 | -0.003 | -0.003 | -0.024 | -0.038 |
| TROP | 0.037 | -0.239 | -0.224 | -0.388 | -0.685 |
| NHml | 0.329 | -1.423 | -1.415 | -1.315 | -0.729 |
| ARCT | 0.171 | -0.859 | -0.488 | -0.413 | -0.143 |
| ERF | | | | | |
| SH | 0.729 | 0.608 | 0.663 | 0.511 | 0.628 |
| TROP | 0.081 | -0.170 | -0.415 | -0.330 | -0.489 |
| NHml | -0.184 | -1.774 | -1.710 | -1.752 | -0.904 |
| ARCT | -0.139 | -1.046 | -0.900 | -1.075 | -0.149 |



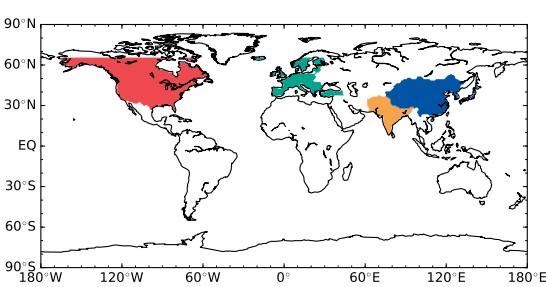

**Figure 1.** Emission regions according to the HTAP definition. The colours represent: green - Europe (EU), red - North America (NA), blue - East Asia (EA) and yellow - South Asia (SA).



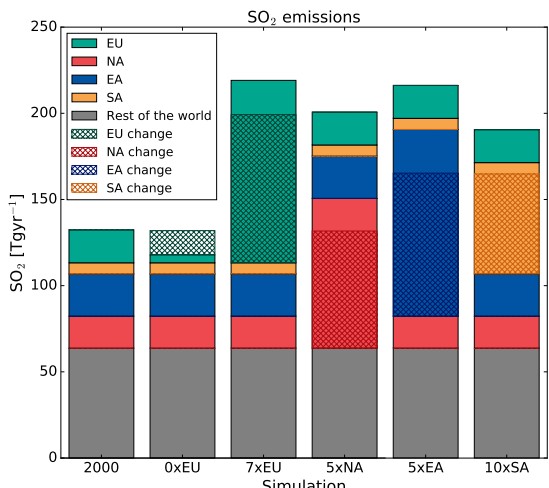

**Figure 2.** Global annual SO₂ and regional emissions and emission differences in the simulations. Each column shows the total global SO₂ emissions in each simulation and the colour shading indicates the contribution from each region. Hatching indicates the emission change relative to the year 2000 simulation.





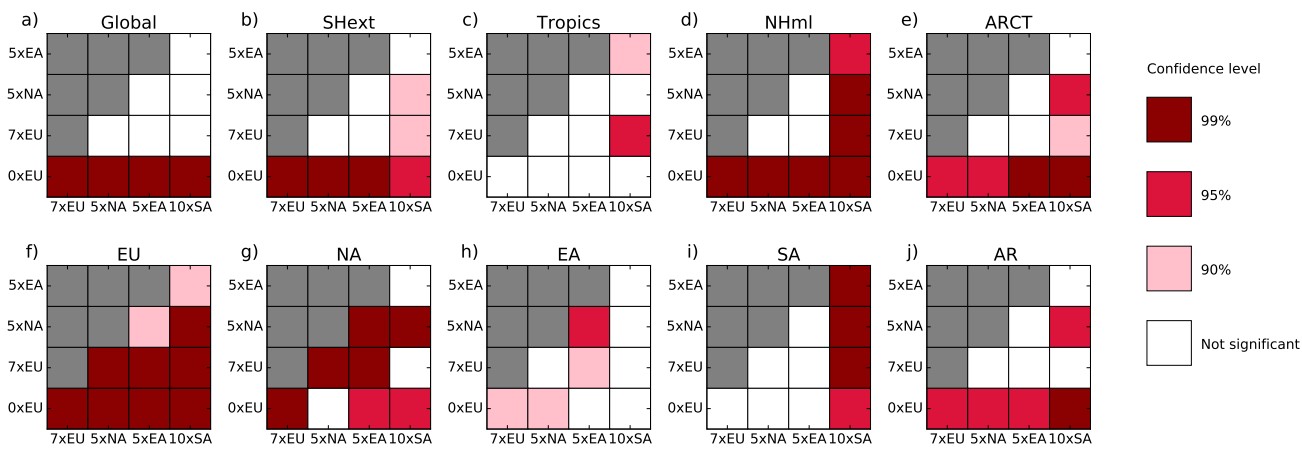

**Figure 3.** Significance levels for temperature differences between the different experiments, for the temperature response regions a) global mean, b) SHext, c) Tropics, d) NHml, e) ARCT, f) EU, g) NA, h) EA, i) SA and j) AR.





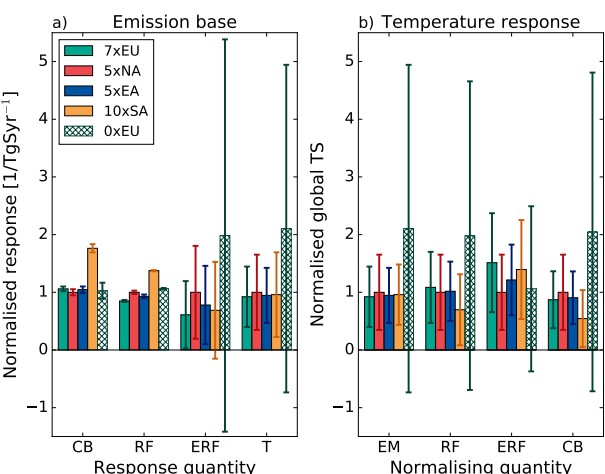

**Figure 4.** Normalised a) column burden (CB), radiative forcing (RF), effective radiative forcing (ERF) and temperature (T) per unit $SO_2$ emission, and b) normalised temperature response per emissions, RF, ERF and CB in the different experiments. The error indicate show one standard deviation.





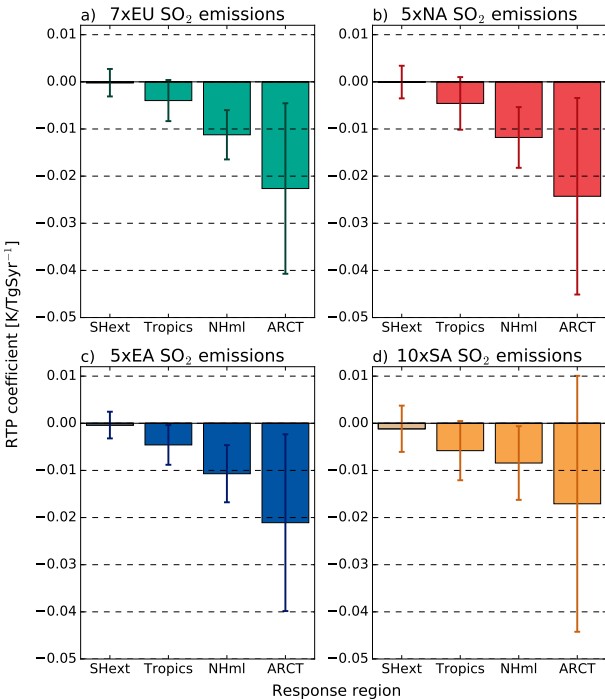

**Figure 5.** Latitudinal RTP coefficients for $SO_2$ emission [K/TgSyr$^{-1}$] for a) EU emissions b) NA emissions c) EA emissions and d) SA emissions. Grey shading indicates that the temperature change is not statistically significant ($p > 0.05$) compared to the control simulation. The error bars indicate one standard deviation.





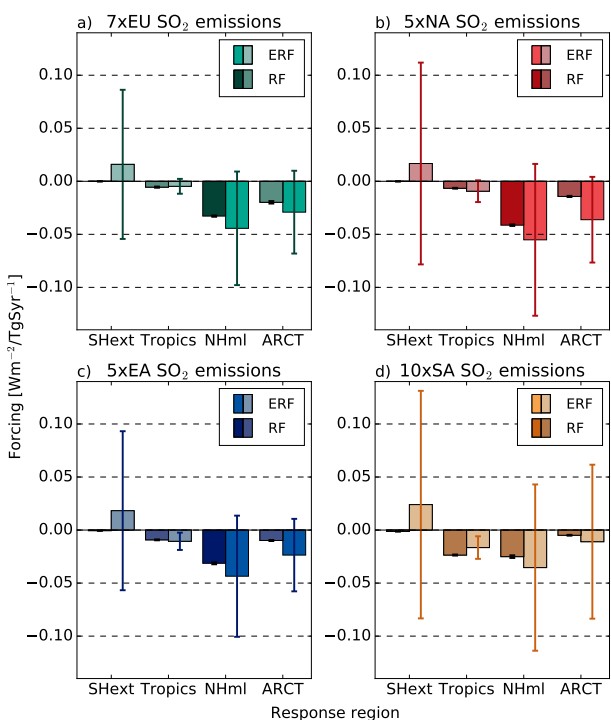

**Figure 6.** Latitudinal RF and ERF for $SO_2$ emission [$Wm^{-2}/TgSyr^{-1}$] for a) EU emissions b) NA emissions c) EA emissions and d) SA emissions. In each pair of bars the left bar indicated RF and the right bar indicated ERF. Grey shading indicates that the forcing response is not statistically significant ($p > 0.05$) compared to the control simulation.

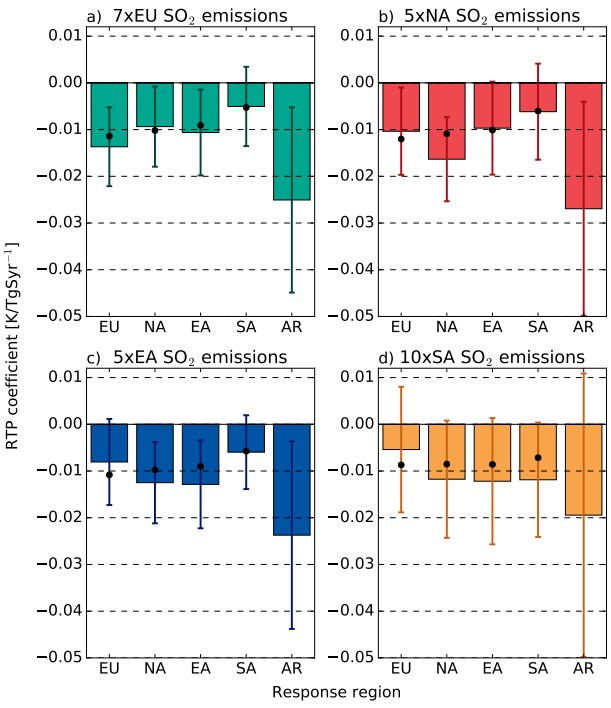

**Figure 7.** Regional RTP coefficients for $SO_2$ emission [K/TgSyr$^{-1}$]for a) EU emissions b) NA emissions c) EA emissions and d) SA emissions. Grey shading indicates that the temperature change is not statistically significant ($p > 0.05$) compared to the control simulation. The error bars indicate one standard deviation. Black dots indicate the zonal mean for the latitudes that cover each region.





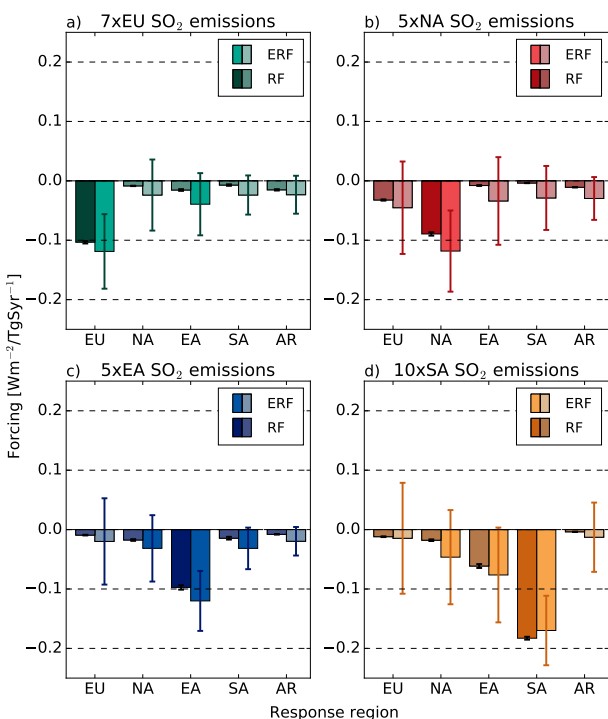

**Figure 8.** Regional RF and ERF for $SO_2$ emission [$Wm^{-2}/TgSyr^{-1}$] for a) EU emissions b) NA emissions c) EA emissions and d) SA emissions. In each pair of bars the left bar indicated RF and the right bar indicated ERF. Grey shading indicates that the forcing response is not statistically significant ($p > 0.05$) compared to the control simulation. The black dots indicate the zonal mean of the latitudes covering each response region.





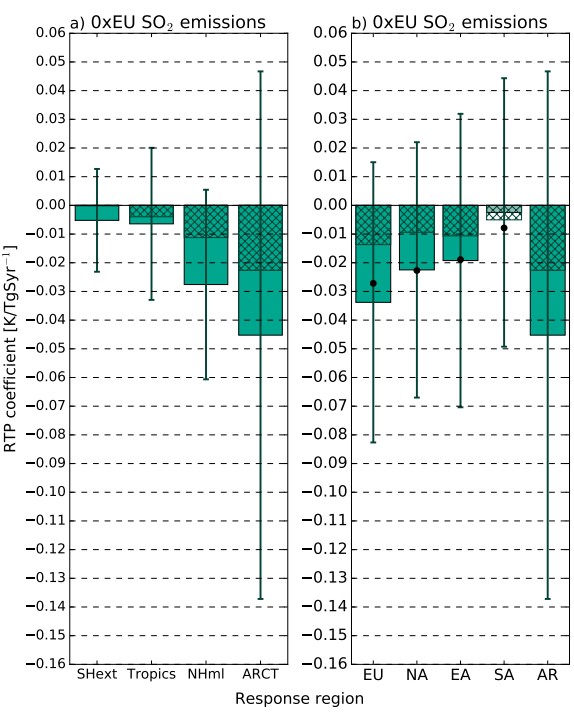

**Figure 9.** Latitudinal (a) and regional (b) RTP coefficients for 0xEU SO$_2$ emissions. [K/TgSyr$^{-1}$]. Grey shading indicates non-statistical differences ($p > 0.05$). The hatching indicated the RTP for 7xEU emissions (cf. Fig. 5 and 7) for easy comparison.





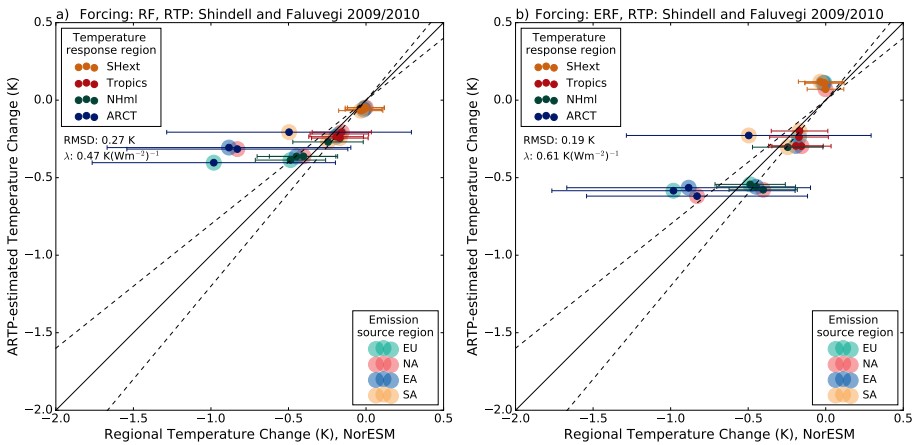

**Figure 10.** Regional temperature change from the coupled simulations (horizontal axis) compared with the estimated temperature response when using a) RF and b) ERF in combination with the RTP coefficients of Shindell and Faluvegi (2009), Eq. 1 with the climate sensitivity derived from the current experiments (vertical axis). The horizontal bars indicate one standard deviation for the temperature response in the coupled simulations. The dashed lines show ±20% agreement threshold.





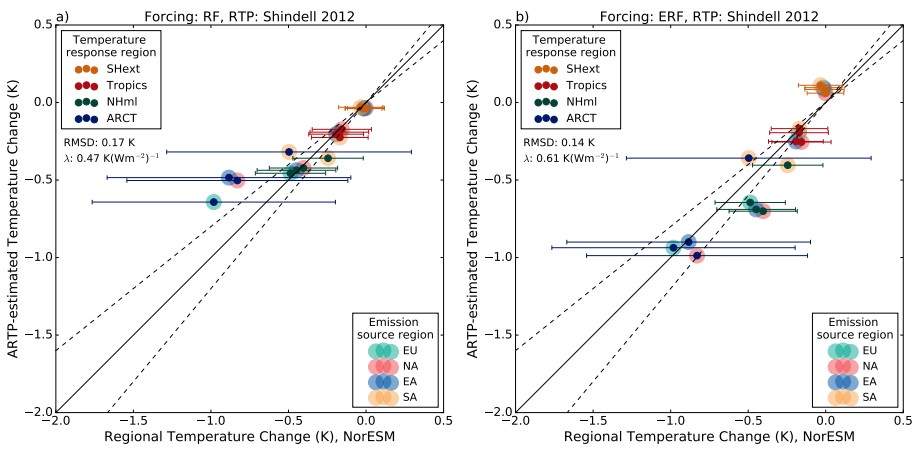

**Figure 11.** As Fig. 10 but with the RTP coefficients of Shindell (2012), Eq. 2 with the climate sensitivity derived from the current experiments.





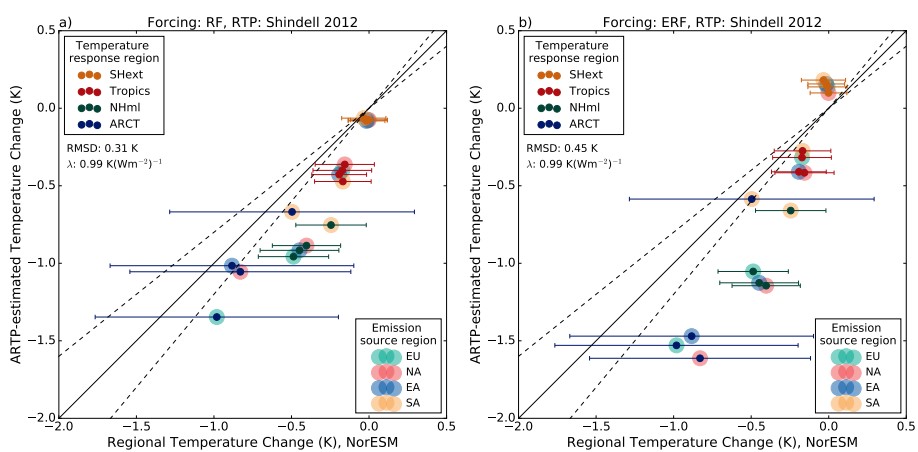

**Figure 12.** As Fig. 10 but with the RTP coefficients of Shindell (2012), Eq. 2 with the $CO_2$ sensitivity from Iversen et al. (2013).



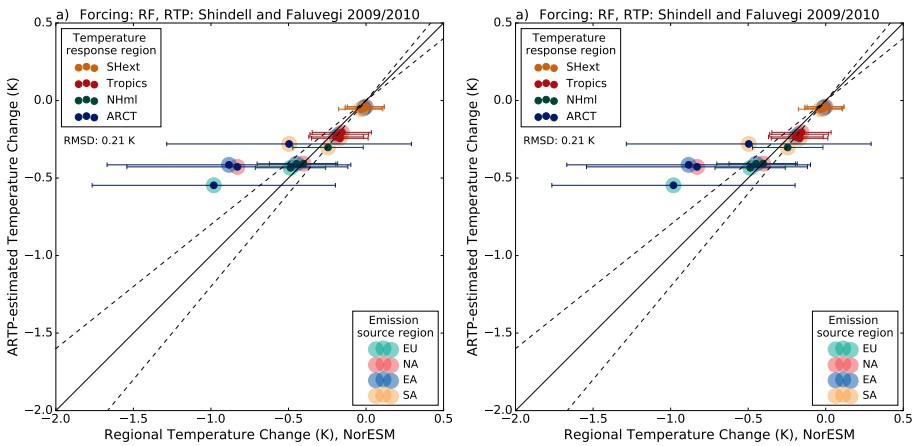

**Figure 13.** As Fig. 10 but with the RTP coefficients of Shindell and Faluvegi (2009), and with no climate sensitivity applied.