# Peer review of "Local and remote temperature response of regional SO2 emissions"

_Atmospheric Chemistry and Physics, 2018_

## Referee Comment (RC1) · Anonymous Referee #1 · 9 Oct 2018

This manuscript presents a very interesting and important set of coupled atmosphere-ocean simulations, in which SO2 emissions have been increased or decreased separately in different geographic regions. Studies which have systematically performed aerosol forcing experiments in different individual regions like this are rare, and so the addition of these simulations with the NorESM model is a very valuable addition to the existing literature, that allows for comparison with other models. This manuscript also goes further by looking at both increasing and decreasing aerosol perturbations, allowing the authors to test the linearity of the response, and showing that there may be considerable non-linearities which is an important caveat when constructing regional metrics.

I recommend the manuscript be accepted provided the following points are addressed:

[Figure]

Scientific comments:

L26/Introduction: The authors should additionally mention the work of Conley et al. (JGR:A 2018, https://doi.org/10.1002/2017JD027411) and Kasoar et al. (npj Climate and Atmospheric Science 2018, https://doi.org/10.1038/s41612-018-0022-z) which are both highly relevant and could be compared directly with this study. Conley et al. present global temperature changes to removing US SO2 emissions in three different atmosphere-ocean models, while Kasoar et al. show temperature responses due to removing SO2 emissions individually from either North America, Europe, South Asia, or East Asia in the HadGEM3 atmosphere-ocean model, directly analogous to the present study with NorESM.

L182: Are the 'climatology' aerosols diagnosed from the control simulation? Or do they come from somewhere else? Essentially, I want to double-check that a free-running control simulation would by construction have zero RF – which might not be the case if the 'climatology' isn't equal to the online control aerosol distribution.

Figure 3: Though a nice way of presenting this info, I'm not sure this figure is critical given that some of the information can also be discerned from the error bars e.g. on Fig. 5. The current Fig. 3 could probably be moved to the supplement. Instead, what would be much more useful to include here would be global maps of the temperature changes in each of the experiments, perhaps with shading or stippling to indicate significance at each grid point. This would allow the same comparison as the present Figure 3 in terms of seeing how similar and significant the responses are in different regions, but would also allow for the pattern of the temperature responses to be compared against other studies. I find it odd that this paper does not currently include a single plot which just shows the geographic temperature changes from these experiments, which to me is the first thing I would look at.

Table 2: It could be useful to also include the climate sensitivity values (dT/ERF and dT/RF) in this table, as they are used later in the discussion.
Table 2: Please include uncertainty ranges

L206/Figure 3 plus subsequent figures that have error bars: How is significance determined? The paper frequently discusses whether results from different regions are significantly different from each other, but I am unclear how this was tested for. Similarly error bars are sometimes quoted as one standard deviation, but in the absence of a large ensemble of simulations I'm unclear what it's a standard deviation of.

L234: Quoting a correlation coefficient for four data points (three of which are pretty much on top of each other) is arguably misleading – it's bound to be close to 1, but this doesn't necessarily tell you much about the strength of the relationship given that not much variation was sampled

L239-249/Figure 4b: This is a very interesting result which I find it hard to get my head around. The whole reason that ERF is widely used is because it has generally been shown to be a better predictor of dT than the instantaneous RF – at least across different and varied forcings. Here you find the opposite – but moreover finding that emission change is an even better predictor of dT! Given that sulfate aerosol has little atmospheric absorption and affects the surface temperatures pretty much entirely through TOA radiative forcing, I would really like to understand why the ERF correlates worse with dT than the emission change. Do the authors have any ideas, physically, how this comes about in these experiments? E.g. maybe some large land-surface responses in the fixed SST experiments, which mean that a substantial portion of the final temperature response is subsumed in rapid adjustments? (N.B. As noted later in the manuscript though, once you include the 0xEU experiment, then ERF does become a better predictor of dT again).

L262: The SA response can't be weaker in all the latitude bands, or else it would also be weaker globally.

L263-L270/Figure 6: I'm confused by the different indications of significance. E.g in Figure 6d, the 10xSA ERF in the tropics has an error bar which does not cross zero,

and yet it is shaded to indicate that it's not significant. Yet in Fig 6b, the 5xNA ERF in the NHml has a huge error bar that spans zero, but is shaded to indicate that it is significant.

L316-318: Consider citing Teng et al. (GRL 2012, doi:10.1029/2012GL051723) which provides a similar example of aerosol forcing over Asia resulting in remote warming over the US, in a different model

L415: Units of climate sensitivity seem to have been inverted here. Check that the number being quoted isn't actually the feedback parameter.

L422: Why is the goal to use model-dependent sensitivities? Surely for integrated assessment modelling, you would like to use a model-independent choice of climate sensitivity? So then, does it matter if you assume a different climate sensitivity and get a different answer (scaled up or scaled down) as a result?

L424: why do you compare with the GISS-E2 transient sensitivity and not equilibrium, given that the NorESM simulations aren't transient?

L427-428: I don't agree with this conclusion. The way I read it, using the Shindell and Faluvegi coefficients has reproduced NorESM well here because GISS-E2 has an ECS (i.e. 2 x CO2) climate sensitivity of 0.6 K/(Wm-2) (Flato et al. IPCC 2013), which happens to be very similar to the sulfate climate sensitivity found for NorESM here. Hansen et al. (JGR 2005, doi:10.1029/2005JD005776) shows that in GISS-E2, the ERF-based sulfate sensitivity is similar to CO2. So this doesn't explain why the authors get such different climate sensitivities for sulfate and 2 x CO2 simulations in NorESM. Maybe due to differences in methodology defining the equilibrium state? Or in calculating ERF (e.g. fixed-SST versus Gregory regression?). At any rate, it would be interesting to understand why NorESM seems to have such a different climate sensitivity for sulfate here compared with the previously published 2 x CO2 values. The message of e.g. the Hansen et al. (2005) paper is that ERF is a more forcing-independent predictor of temperature change, so it's surprising that the global climate

sensitivity of NorESM varies so much between forcers. One final point, is that climate sensitivities in general differ hugely between models for the same forcing agent (e.g. the range in Flato et al. is from ∼0.5 to 1.5 K/(Wm-2)). This is presumably the case for sulfate as much as any other forcer. So, the coincidence that GISS-E2 has a similar climate sensitivity to NorESM doesn't really show that there is smaller variation across models in the sulfate climate sensitivity compared with between different forcers in the same model; this seems quite unlikely to be the case across most models in fact.

L486-487: If saturation of aerosol indirect effects is the explanation here, then shouldn't there be a similar difference in the RF/em as there is in the ERF/em? I don't see how the RF/em can be unaffected by CDNC saturation such that it only shows up as a difference in ERF/em. On a related point: In Figure 4, the error bars for the 0xEU response per em or per RF are enormous and span the entire range of the other experiments. Can the authors be confident that the sensitivity to an emissions reduction actually is any different to an emissions increase, given the considerable overlap of the error bars? It might just be that the smaller forcing and smaller response from the 0x experiment has higher uncertainty because the signal is small compared to internal model variability.

Technical/grammar/typographic comments:

L69-70: Confusing wording in this sentence, please re-phrase

L176-177 and Eq3: Inconsistent use of r subscript (emission region or response region?)

L217: add 'typically' or similar caveat

Figure 4: The caption should explain how the quantities are normalised. Currently, have to refer to the main text to find out that everything is normalised to the 5xNA experiment in this plot. L263: Should Figure 6 have been referenced here?

L299: increase -> increases

L495: skills -> skill

---

## Referee Comment (RC2) · Anonymous Referee #2 · 12 Oct 2018

The authors conduct a variety of regional SO2 emissions perturbation simulations in NorESM to calculate emissions-temperature metrics for several world regions. They increase emissions in each region by a factor that results in an equivalent global mean radiative forcing change, and decrease emissions to zero over Europe only to test the non-linearity of temperature response to SO2 emissions. Overall, the methods are mostly sound and the results are interesting. I recommend publication subject to minor revisions.

Main comments:

1) It is worth citing and mentioning Conley et al. (2018), which looks at climate response to removal of US SO2 emissions. There are some possibilities for comparison and discussion, such as their Table 3 which includes an estimate of temperature re-

sponse per unit emission change of SO2.

2) Global maps of the temperature response to each of these SO2 perturbations would strengthen this paper. Likewise I think global maps of ERF would be interesting as well. This would put the results in context of the few multimodel studies on this topic, such as the one cited above and Kasoar et al. (2016) which you mention already in the manuscript.

3) The biggest weakness of the paper is the use of a single coupled climate model, especially in a time when multimodel studies are becoming the norm. Ideally, the emissions-based RTP coefficients could be based on an average of several disparate models for more robustness. I understand that it's not feasible to do that in this study, but perhaps the authors could comment on whether or not they expect their results to be robust across additional CMIP models?

4) In Fig 4a and b, the error bars for just one standard deviation from the mean are quite large for the zero EU SO2 emissions perturbation How can the authors then be so sure about a nonlinearity in the response depending on the magnitude and sign of the emissions changes? Since the zero-out EU SO2 emissions perturbation is much smaller in absolute magnitude than the 7xEU, you would likely need a slightly longer simulation than 160 years to reduce those error bars. Otherwise, I'm not sure how you can rule out the role of internal climate variability.

Other minor comments: 1) I'm not really seeing the grey shading in Figure 5? Is it there but just really small?

2) L332-333: this isn't a complete sentence. In general I think the phrase "e.g." is overused in this manuscript and seems to be rather unconventional to start sentences with that abbreviation which happens a couple of times here.

3) L84: "an comparison" should be "a comparison"

---

## Author Comment (AC1) · 1 Dec 2018

Response to reviewers

We thank the two reviewers for their constructive comments and helpful suggestions. Below are our responses to each comment.

Reviewier #1

Scientific comments:

Comment:

L26/Introduction:

[Figure]

The authors should additionally mention the work of Conley et al. (JGR:A 2018, https://doi.org/10.1002/2017JD027411) and Kasoar et al. (npj Climate and Atmospheric Science 2018, https://doi.org/10.1038/s41612-018-0022-z) which are both highly relevant and could be compared directly with this study. Conley et al. present global temperature changes to removing US SO2 emissions in three different atmosphere-ocean models, while Kasoar et al. show temperature responses due to re- moving SO2 emissions individually from either North America, Europe, South Asia, or East Asia in the HadGEM3 atmosphere-ocean model, directly analogous to the present study with NorESM.

Response: We thank the reviewer for the suggestion. References to Conley et al (2018) and Kasoar et al. (2018) have been added to the manuscript, line 51, 59 and in the Discussion as well as in Summary and Conclusions.

Change in manuscript: "Differences between RTP coefficients derived from different climate models can stem from a number of different sources, involving everything from atmospheric processing of aerosols, interaction with radiation, aerosol cloud effects or climate feedbacks, and how these processes are represented in different climate models (Kasoar et al., 2016; Conley et al., 2018)."

Comment:

L182: Are the 'climatology' aerosols diagnosed from the control simulation? Or do they come from somewhere else? Essentially, I want to double-check that a free-running control simulation would by construction have zero RF – which might not be the case if the 'climatology' isn't equal to the online control aerosol distribution.

Response: The climatological aerosols are in this case the native CAM4 aerosols. However, the fixed SST simulations with dual calls were performed for the reference year 2000 aerosol emissions. Thus, the RF is derived from the difference in radiation between two fixed SST simulations: one with year 2000 aerosol emissions and one with the perturbation emissions, and both with meteorology defined by the native CAM4

aerosol. Thus, the RF is by definition zero in the control experiment, to the extent that the atmospheric aerosol loading resulting from the CAM-Oslo aerosol scheme is similar in the fixed SST simulation and the coupled simulation. We agree that the description of the RF simulations is not clear and we have rewritten this sections: line 184-190.

Change in manuscript: "The RF is derived from fixed Sea Surface Temperatures (SSTs) simulations where dual calls are made to the radiation code: one call with the CAM4 climatological aerosols and another call where the emission perturbation aerosol concentrations and their effect on cloud albedo are sent to the radiation code solely for diagnosing the radiative effect of these. Thus the meteorology in the RF simulations is identical since the radiative effects of the emission perturbations do not feedback on the meteorology. Similarly, a dual call control simulation with year 2000 aerosol emissions was performed. With this methodology the radiative effects alone from the aerosol can be quantified, without influence of fast or slow feedbacks. The RF simulations are 7 years long and the 5 last years are used for the analysis."

Comment:

Figure 3: Though a nice way of presenting this info, I'm not sure this figure is critical given that some of the information can also be discerned from the error bars e.g. on Fig. 5. The current Fig. 3 could probably be moved to the supplement. Instead, what would be much more useful to include here would be global maps of the temperature changes in each of the experiments, perhaps with shading or stippling to indicate significance at each grid point. This would allow the same comparison as the present Figure 3 in terms of seeing how similar and significant the responses are in different regions, but would also allow for the pattern of the temperature responses to be com- pared against other studies. I find it odd that this paper does not currently include a single plot which just shows the geographic temperature changes from these experiments, which to me is the first thing I would look at.

Response: As suggested by the reviewer, Figure 3 has been moved to the supplementary material, and global maps of the temperature response and ERF has been added, also by request from reviewer #2. Please see Figures 5 and 7 in the revised manuscript.

Comment:

Table 2: It could be useful to also include the climate sensitivity values (dT/ERF and dT/RF) in this table, as they are used later in the discussion. Table 2: Please include uncertainty ranges

Response: We agree with the reviewer that it would be useful to include the climate sensitivity values. We have added these values (dT/RF and dT/ERF) to Table 2 as well as standard deviations. Please see Table 2.

Comment:

L206/Figure3 plus subsequent figures that have error bars: How is significance determined? The paper frequently discusses whether results from different regions are significantly different from each other, but I am unclear how this was tested for. Similarly error bars are sometimes quoted as one standard deviation, but in the absence of a large ensemble of simulations I'm unclear what it's a standard deviation of.

Response: The students's t –test is used to determine if the changes are statistically significant. The standard deviations represent the variability of the data from where the mean value is extracted, i.e. from each simulation. This information has been added to the manuscript, line 166 -167 and 195-197.

Change in manuscript: "All the results presented are annual mean quantities and the first 50 years of each simulations have been removed before averaging and are tested for statistical significance with a student's t-test. Uncertainty ranges for the results are given as standard deviation representing the variability within the experiments."

Comment:

L234: Quoting a correlation coefficient for four data points (three of which are pretty much on top of each other) is arguably misleading – it's bound to be close to 1, but this doesn't necessarily tell you much about the strength of the relationship given that not much variation was sampled.

Response: The point is well taken, the correlation has been removed.

Comment:

L239-249/Figure 4b: This is a very interesting result which I find it hard to get my head around. The whole reason that ERF is widely used is because it has generally been shown to be a better predictor of dT than the instantaneous RF – at least across different and varied forcings. Here you find the opposite – but moreover finding that emission change is an even better predictor of dT! Given that sulfate aerosol has little atmospheric absorption and affects the surface temperatures pretty much entirely through TOA radiative forcing, I would really like to understand why the ERF correlates worse with dT than the emission change. Do the authors have any ideas, physically, how this comes about in these experiments? E.g. maybe some large land-surface responses in the fixed SST experiments, which mean that a substantial portion of the final temperature response is subsumed in rapid adjustments? (N.B. As noted later in the manuscript though, once you include the 0xEU experiment, then ERF does become a better predictor of dT again).

Response: We were also surprised by this result, and we have to admit that we have no definite explanation at this stage. We agree that it would indeed be interesting to better understand why the emission changes predicts the temperature response so well in the emission increase experiments, but a detailed investigation of this is outside the scope of the current study. Our aim with this study is primarily to provide and evaluate coefficients that are easy to use in the context of Integrated Assessment Modelling, and thus we have avoided in depth discussions about the underlying physical mechanisms. Our conjecture at the moment is that this is a manifestation of saturation of aerosol indirect effects which would render the relationship between emission and temperature more linear when emissions continue to increase. This would potentially also explain why RF predicts the global temperature increase well for the emission increase experiments. The result, as the reviewer points out, is only apparent in the experiment where emissions were increased. This is, however, something we would like to study in more detail in the future. This is discussed in the manuscript, line 499-504.

Comment:

L262: The SA response can't be weaker in all the latitude bands, or else it would also be weaker globally.

Response: We thank the reviewer for pointing out this inconsistency in the description if the result. This sentence has been rephrased to clarify that the response in 10xSA is weaker in NHml and ARCT and stronger in SHext and Tropics compared to the other experiments, line 268-270.

Change in manuscript: "Thus, the latitudinal temperature responses are in principle indistinguishable for emission increases from EU, NA and EA, while the SA emission response is weaker in NHml and ARCT while it is stronger in SHext and Tropics compared to the other experiments. The spatial distributions of the temperature responses are shown in Figure 5."

Comment:

L263-L270/Figure 6: I'm confused by the different indications of significance. E.g in Figure 6d, the 10xSA ERF in the tropics has an error bar which does not cross zero, and yet it is shaded to indicate that it's not significant. Yet in Fig 6b, the 5xNA ERF in the NHml has a huge error bar that spans zero, but is shaded to indicate that it is significant.

Response: We have examined the significance for these results looking at the definition of the student's t-test. Here, the fact that the magnitude of the difference between

the 5xNA and control simulation is larger than 10xSA and control simulations is the key factor. The absolute difference between the mean values has a larger influence than the difference in the variances in this case for determining if the result is statistically significant. However, the 10xSA result is close to being significant at the 95% confidence level with p = 0.065.

Comment:

L316-318: Consider citing Teng et al. (GRL 2012, doi:10.1029/2012GL051723) which provides a similar example of aerosol forcing over Asia resulting in remote warming over the US, in a different model.

Response: We thank the reviewer for the reference suggestion. A reference to Teng et al. (2012) has been added, line 544.

Change in manuscript: "Moreover, Teng et al. (2012) found a temperature impact in North America directly linked to absorbing aerosols in Asia."

Comment:

L415: Units of climate sensitivity seem to have been inverted here. Check that the number being quoted isn't actually the feedback parameter.

Response: The number given is indeed a feedback parameter, or climate response, and the value used for the calculations made in the manuscript was the inverted value. The reason for choosing to present the value as a feedback parameter was for transparency with respect to Iversen et al. (2013), Table 1, from where the number is taken. In Iversen et al. (2013), different climate sensitivity estimates are presented as feedback parameters with the units Wm2/K. We noticed, however, that the value had inadvertently been miscited and should read 1.101, not 1.01 Wm2/K. The calculations and figures have been corrected and the conclusions are not affected by this mistake. We have rephrased this paragraph so that it should be clear for the reader that the value cited is the climate sensitivity, line 426.

Change in manuscript: "For NorESM this climate sensitivity has been estimated to 0.91 K/Wm $-2$ (Iversen et al., 2013, $\lambda$ reg in Table 1)."

Comment:

L422: Why is the goal to use model-dependent sensitivities? Surely for integrated assessment modelling, you would like to use a model-independent choice of climate sensitivity? So then, does it matter if you assume a different climate sensitivity and get a different answer (scaled up or scaled down) as a result?

Response: The point we would like to make in that paragraph is that there is no model-independent sensitivity or RTP coefficient. We see that this was not clear in the manuscript and have rephrased parts of this paragraph, line 435-440. Even if no climate sensitivity is explicitly applied, there is inherently the climate sensitivity of the model simulations from which the RTP coefficients were derived integrated in the co-efficients. The equilibrium sensitivity of the GISS model can only be used by explicitly applying that parameter in the equations for the ARTP. What we wanted to show is how well the RTP coefficients of Shindell and Faluvegi (2010) work without weighting with a climate sensitivity, which means that the sensitivity of that particular model simulation is what is, in effect, used. If it matters what sensitivity is applied in the equation will depend on what the RTP coefficients will be used for. When comparing the effect of different forcing agents in integrated assessment modelling, it would be crucial, but when comparing the relative latitudinal temperature change distribution, it could be considered less important. However, an assumption about the climate sensitivity always has to be made no matter what the application is, which is what we would like to emphasise. However, an uncertainty related to the RTPs are useful for a sensitivity analysis in the integrated assessment analysis.

Change in manuscript: "A third alternative is to apply the RTP coefficients without normalising with a model dependent climate sensitivity parameter, i.e. using the RTP-coefficients of Shindell and Faluvegi (2010), Table 3, directly with forcing estimates

[Figure]

(Fig. 14). The implicit assumption in this method is that the sensitivity of NorESM to aerosol forcing is equal to that of the GISS model simulations used to derive the RTP coefficient. This is is equivalent with applying the GISS model's sensitivity of 0.5 K/Wm $-2$ (Shindell, 2012) in Equation 2. This assumption about the sensitivity leads to RTP-derived temperature responses with smaller RMSD values than both those derived by applying the ECS for NorESM in Equation 1 and 2."

Comment:

L424: why do you compare with the GISS-E2 transient sensitivity and not equilibrium, given that the NorESM simulations aren't transient?

Response: The climate sensitivity is an integral part of the formulation of the RTP-coefficients of Shindell and Faluvegi (2010). If a climate sensitivity is not explicitly applied, the implication is that the climate sensitivity from the simulation the RTP coefficients is used. This sensitivity is equal to the transient climate sensitivity of the GISS model (Shindell, 2012). We have rewritten this paragraph to make this clearer for the reader, line 435-440. Please also see discussion above.

Comment:

L427-428: I don't agree with this conclusion. The way I read it, using the Shindell and Faluvegi coefficients has reproduced NorESM well here because GISS-E2 has an ECS (i.e. 2 x CO2) climate sensitivity of 0.6 K/(Wm-2) (Flato et al. IPCC 2013), which happens to be very similar to the sulfate climate sensitivity found for NorESM here. Hansen et al. (JGR 2005, doi:10.1029/2005JD005776) shows that in GISS-E2, the ERF-based sulfate sensitivity is similar to CO2. So this doesn't explain why the authors get such different climate sensitivities for sulfate and 2 x CO2 simulations in NorESM. Maybe due to differences in methodology defining the equilibrium state? Or in calculating ERF (e.g. fixed-SST versus Gregory regression?). At any rate, it would be interesting to understand why NorESM seems to have such a different climate sensitivity for sulfate here compared with the previously published 2 x CO2 values.

The message of e.g. the Hansen et al. (2005) paper is that ERF is a more forcing-independent predictor of temperature change, so it's surprising that the global climate sensitivity of NorESM varies so much between forcers. One final point, is that climate sensitivities in general differ hugely between models for the same forcing agent (e.g. the range in Flato et al. is from âĹij0.5 to 1.5 K/(Wm-2)). This is presumably the case for sulfate as much as any other forcer. So, the coincidence that GISS-E2 has a similar climate sensitivity to NorESM doesn't really show that there is smaller variation across models in the sulfate climate sensitivity compared with between different forcers in the same model; this seems quite unlikely to be the case across most models in fact.

Response: The point is well taken, this sentence has been removed. We have also clarified that the climate sensitivity derived from the experiments presented here is not directly comparable to the equilibrium climate sensitivity, line 427-429.

Change in manuscript: "This is higher than the sensitivity to aerosol forcing obtained in this study. The climate sensitivity from the simulations presented here is not directly comparable with an equilibrium climate sensitivity, since an equilibrium temperature response would require considerably longer simulations for allowing the ocean to fully adjust."

Comment:

L486-487: If saturation of aerosol indirect effects is the explanation here, then shouldn't there be a similar difference in the RF/em as there is in the ERF/em? I don't see how the RF/em can be unaffected by CDNC saturation such that it only shows up as a difference in ERF/em. On a related point: In Figure 4, the error bars for the 0xEU response per em or per RF are enormous and span the entire range of the other experiments. Can the authors be confident that the sensitivity to an emissions reduction actually is any different to an emissions increase, given the considerable overlap of the error bars? It might just be that the smaller forcing and smaller response from the 0x experiment has higher uncertainty because the signal is small compared to internal model variability.

Response: There is a larger RF response per unit emission for 0xEU compared to 7xEU (Figure 3). In the RF simulations the cloud albedo effect is diagnosed from the cloud distribution determined by the fixed meteorology. Thus, the indirect effect is restricted by the cloud distribution and represent only this first indirect effect (constant cloud water content but changed CDNC). In the ERF simulations the cloud distribution and liquid water content respond to the changed aerosol. In particular, the liquid water path shows large changes which impacts the opacity of the clouds. Thus, including the life-time effect and semidirect effects on clouds amplifies the indirect effects in the ERF simulations compared to the RF simulations. The results are certainly associated with a lot of uncertainties, as discussed in the manuscript. However, the fact that the results are normalised by the emission change inflates the standard deviation of the 0xEU experiment compared to the other experiments. In fact, the variability is similar in all simulations, see Figure 1 below. The temperature responses in the different experiments have also been tested for significance with the student's t-test, and the global temperature response to European SO2 emission reductions is found to be statistically different to the temperature responses in emission increase experiments, see Figure S1a. We have now clarified this in the manuscript, line 335-337. We have also emphasised in the text that uncertainties are large and that what we present is no proof of nonlinearity, but an indication that there is no reason to assume that the relationship between emission change and temperature response would be linear, line 562-564.

Change in manuscript: "Furthermore, indications were found that the emission-based RTPs derived with NorESM might be non-linear. Removal of anthropogenic European SO 2 emissions led to a temperature response per unit emission approximately twice of that in the 7xEU experiment in NorESM. The result is, however, associated with large uncertainties."

Technical/grammar/typographic comments:

Comment:

L69-70: Confusing wording in this sentence, please re-phrase

Response: This sentence has been rephrased, line 70-71.

Change in manuscript: "Lately, the radiative forcing of long-lived greenhouse gases other than CO 2 have been included in GAINS, which makes it possible to evaluate the changes emissions of these due to air pollution abatement."

Comment:

L176-177 and Eq3: Inconsistent use of r subscript (emission region or response region?)

Response: The subscripts have been changed for equation 3 to be consistent with equations 1 and 2.

Comment:

L217: add 'typically' or similar caveat

Response: It has been clarified that this results pertain to the experiments we present in this study, line 224-225.

Change in manuscript: "Thus, on a global scale, fast cloud feedbacks contribute to dampen the forcing effect of the emission increases in the NorESM experiments presented here."

Comment:

Figure 4: The caption should explain how the quantities are normalised. Currently, have to refer to the main text to find out that everything is normalised to the 5xNA experiment in this plot.

Response: We agree with the reviewer. An explanation has been added to the figure caption of Figure 3.

Comment:

263: Should Figure 6 have been referenced here?

Response: We thank the reviewer for noticing this mistake. A reference to Figure 6 has been added here, line 272.

Change in manuscript: "The only latitudinal RF and ERF that are statistically significant are the responses to emissions increases in EU, NA and EA, in NHml, the latitudinal band inside which these emission regions are located (Fig. 6)."

Comment:

L299: increase -> increases

Response: The mistake has been corrected, line 307.

Change in manuscript: "The column burden increases by..."

Comment:

L495: skills -> skill

Response: We thank the reviewer for noticing. This has been corrected, line 508.

Change in manuscript: "...have the best skill to predict the temperature response."

Reviewer #2

Main comments:

Comment:

1) It is worth citing and mentioning Conley et al. (2018), which looks at climate response to removal of US SO2 emissions. There are some possibilities for comparison and discussion, such as their Table 3 which includes an estimate of temperature response per unit emission change of SO2.

Response: We thank the reviewer for the suggestion. We have now included a citation of Conley et al. (2018) in the manuscript, line 52 and in the Discussion as well as in Summary and Conclusions

Change in manuscript: "Differences between RTP coefficients derived from different climate models can stem from a number of different sources, involving everything from atmospheric processing of aerosols, interaction with radiation, aerosol cloud effects or climate feedbacks, and how these processes are represented in different climate models (Kasoar et al., 2016; Conley et al., 2018)."

Comment:

2) Global maps of the temperature response to each of these SO2 perturbations would strengthen this paper. Likewise I think global maps of ERF would be interesting as well. This would put the results in context of the few multimodel studies on this topic, such as the one cited above and Kasoar et al. (2016) which you mention already in the manuscript.

Response: As advised by both reviewer #1 and #2, global maps of the temperature response and the ERF are now added to the manuscript, please see Figure 5 and 7.

Comment:

3) The biggest weakness of the paper is the use of a single coupled climate model, especially in a time when multimodel studies are becoming the norm. Ideally, the emissions-based RTP coefficients could be based on an average of several disparate models for more robustness. I understand that it's not feasible to do that in this study, but perhaps the authors could comment on whether or not they expect their results to be robust across additional CMIP models?

Response: We have added a comparison with the results of Conley et al. (2018) and Kasoar et al. (2018), and we do find similarities between the results. The strong Arctic response is present in these two studies as well, line 534-537.

Change in manuscript: "These results are in line with those of Conley et al. (2018), who found a similar latitudinal temperature change distribution in three different models in response to removal of US SO 2 emissions, and Kasoar et al. (2018) who conducted a single model study where they found that the Arctic warmed most in response to removal of SO 2 emissions in different regions."

Comment:

4) In Fig 4a and b, the error bars for just one standard deviation from the mean are quite large for the zero EU SO2 emissions perturbation How can the authors then be so sure about a nonlinearity in the response depending on the magnitude and sign of the emissions changes? Since the zero-out EU SO2 emissions perturbation is much smaller in absolute magnitude than the 7xEU, you would likely need a slightly longer simulation than 160 years to reduce those error bars. Otherwise, I'm not sure how you can rule out the role of internal climate variability.

Response: Indeed, the 0xEU simulation (and control simulation) is 200 years (line 165). However, the magnitude of the variability is not larger in the 0xEU simulation compared to the other simulations (please also cf. answer to reviewer #1 and Figure 1 above). The non-normalised variability is similar in all simulations. We see that the way of presenting the result as normalised quantities obscures this fact and we have added a clarification in the manuscript, line 335-336, and emphasised that the result is associated with uncertainties, line 562-564.

Change in manuscript: "Furthermore, indications were found that the emission-based RTPs derived with NorESM might be non-linear. Removal of anthropogenic European SO 2 emissions led to a temperature response per unit emission approximately twice of that in the 7xEU experiment in NorESM. The result is, however, associated with large uncertainties."

Minor comments

Comment:

1) I'm not really seeing the grey shading in Figure 5? Is it there but just really small?

Response: The shading colour in what previously was Figure 5, now Figure 4, has been adjusted to appear more grey. Note that it is only the SHext bars in Figure 4 that are shaded grey.

Comment:

2) L332-333: this isn't a complete sentence. In general I think the phrase "e.g." is overused in this manuscript and seems to be rather unconventional to start sentences with that abbreviation which happens a couple of times here.

Response: E.g. has been changed to "For example", line 343. We have gone through the manuscript with particular attention to this phrase and exchanged with equivalent phrase to make the text more varied.

Change in manuscript: "For example, in extreme conditions the partitioning between different aerosol microphysical paths might change, like condensation and nucleation rates of sulphate (Stier et al., 2006)."

Comment:

3) L84: "an comparison" should be "a comparison"

Response: "An" has been changed to "a", line 86. We thank the reviewer for noticing the mistake.

Change in manuscript: "Last in the Result section is a comparison of the performance..."

———————————————

Fig. 1.

**Supplement:**

[revised manuscript text omitted]

---

## Author Response (AR1)

**Response to reviewers**

We thank the two reviewers for their constructive comments and helpful suggestions. Below are our responses to each comment.

**Reviewier #1**

**Scientific comments:**

**L26/Introduction:**

**The authors should additionally mention the work of Conley et al. (JGR:A 2018, https://doi.org/10.1002/2017JD027411) and Kasoar et al. (npj Climate and Atmospheric Science 2018, https://doi.org/10.1038/s41612-018-0022-z) which are both highly relevant and could be compared directly with this study. Conley et al. present global temperature changes to removing US SO2 emissions in three different atmosphere-ocean models, while Kasoar et al. show temperature responses due to re- moving SO2 emissions individually from either North America, Europe, South Asia, or East Asia in the HadGEM3 atmosphere-ocean model, directly analogous to the present study with NorESM.**

We thank the reviewer for the suggestion. References to Conley et al (2018) and Kasoar et al. (2018) have been added to the manuscript, line 51, 59 and in the Discussion as well as in Summary and Conclusions.

*"…and how these processes are represented in different climate models (Kasoar et al., 2016; Conley et al., 2018)."*

**L182:**

**Are the 'climatology' aerosols diagnosed from the control simulation? Or do they come from somewhere else? Essentially, I want to double-check that a free-running control simulation would by construction have zero RF – which might not be the case if the 'climatology' isn't equal to the online control aerosol distribution.**

The climatological aerosols are in this case the native CAM4 aerosols. However, the fixed SST simulations with dual calls were performed for the reference year 2000 aerosol emissions. Thus, the RF is derived from the difference in radiation between two fixed SST simulations: one with year 2000 aerosol emissions and one with the perturbation emissions, and both with meteorology defined by the native CAM4 aerosol. Thus, the RF is by definition zero in the control experiment, to the extent that the atmospheric aerosol loading resulting from the CAM-Oslo aerosol scheme is similar in the fixed SST simulation and the coupled simulation. We agree that the description of the RF simulations is not clear and we have rewritten this sections: line 184-190.

*"The RF is derived from fixed Sea Surface Temperatures (SSTs) simulations where dual calls are made to the radiation code: one call with the CAM4 climatological aerosols and another call where the emission perturbation aerosol concentrations and their effect on cloud albedo are sent to the radiation code solely for diagnosing the radiative effect of these. Thus the meteorology in the RF simulations is identical since the radiative effects of the emission perturbations do not feedback on the meteorology. Similarly, a dual call control simulation with year 2000 aerosol emissions was performed. With this methodology the radiative effects alone from the aerosol can be quantified,*

*without influence of fast or slow feedbacks. The RF simulations are 7 years long and the 5 last years are used for the analysis."*

**Figure 3:**

**Though a nice way of presenting this info, I'm not sure this figure is critical given that some of the information can also be discerned from the error bars e.g. on Fig. 5. The current Fig. 3 could probably be moved to the supplement. Instead, what would be much more useful to include here would be global maps of the temperature changes in each of the experiments, perhaps with shading or stippling to indicate significance at each grid point. This would allow the same comparison as the present Figure 3 in terms of seeing how similar and significant the responses are in different regions, but would also allow for the pattern of the temperature responses to be compared against other studies. I find it odd that this paper does not currently include a single plot which just shows the geographic temperature changes from these experiments, which to me is the first thing I would look at.**

As suggested by the reviewer, Figure 3 has been moved to the supplementary material, and global maps of the temperature response and ERF has been added, also by request from reviewer #2. Please see Figures 5 and 7 in the revised manuscript.

**Table 2:**

**It could be useful to also include the climate sensitivity values (dT/ERF and dT/RF) in this table, as they are used later in the discussion.**

**Table 2: Please include uncertainty ranges**

We agree with the reviewer that it would be useful to include the climate sensitivity values. We have added these values (dT/RF and dT/ERF) to Table 2 as well as standard deviations. Please see Table 2.

**L206/Figure3 plus subsequent figures that have error bars:**

**How is significance determined? The paper frequently discusses whether results from different regions are significantly different from each other, but I am unclear how this was tested for. Similarly error bars are sometimes quoted as one standard deviation, but in the absence of a large ensemble of simulations I'm unclear what it's a standard deviation of.**

The students's t –test is used to determine if the changes are statistically significant. The standard deviations represent the variability of the data from where the mean value is extracted, i.e. from the time series from each simulation. However, in the revised manuscript we have chosen to use the standard error as an uncertainty indication instead of the standard deviation, except for in Table 2. for in Table 2. This information has been added to the manuscript, line 166 -167 and 195-197.

*"All the results presented are annual mean quantities and the first 50 years of each simulations have been removed before averaging and are tested for statistical significance with a student's t-test. Uncertainty ranges for the results are given as standard errors or standard deviations derived from the variability in each simulation."*

**L234:**

**Quoting a correlation coefficient for four data points (three of which are pretty much on top of each other) is arguably misleading – it's bound to be close to 1, but this doesn't necessarily tell you much about the strength of the relationship given that not much variation was sampled.**

The point is well taken, the correlation has been removed.

**L239-249/Figure 4b:**

**This is a very interesting result which I find it hard to get my head around. The whole reason that ERF is widely used is because it has generally been shown to be a better predictor of dT than the instantaneous RF – at least across different and varied forcings. Here you find the opposite – but moreover finding that emission change is an even better predictor of dT! Given that sulfate aerosol has little atmospheric absorption and affects the surface temperatures pretty much entirely through TOA radiative forcing, I would really like to understand why the ERF correlates worse with dT than the emission change. Do the authors have any ideas, physically, how this comes about in these experiments? E.g. maybe some large land-surface responses in the fixed SST experiments, which mean that a substantial portion of the final temperature response is subsumed in rapid adjustments? (N.B. As noted later in the manuscript though, once you include the 0xEU experiment, then ERF does become a better predictor of dT again).**

We were also surprised by this result, and we have to admit that we have no definite explanation at this stage. We agree that it would indeed be interesting to better understand why the emission changes predicts the temperature response so well in the emission increase experiments, but a detailed investigation of this is outside the scope of the current study. Our aim with this study is primarily to provide and evaluate coefficients that are easy to use in the context of Integrated Assessment Modelling, and thus we have avoided in depth discussions about the underlying physical mechanisms. Our conjecture at the moment is that this is a manifestation of saturation of aerosol indirect effects which would render the relationship between emission and temperature more linear when emissions continue to increase. This would potentially also explain why RF predicts the global temperature increase well for the emission increase experiments. The result, as the reviewer points out, is only apparent in the experiment where emissions were increased. This is, however, something we would like to study in more detail in the future. This is discussed in the manuscript, line 498-503.

**L262:**

**The SA response can't be weaker in all the latitude bands, or else it would also be weaker globally.**

We thank the reviewer for pointing out this inconsistency in the description if the result. This sentence has been rephrased to clarify that the response in 10xSA is weaker in NHml and ARCT and stronger in SHext and Tropics compared to the other experiments, line 268-270.

*"Thus, the latitudinal temperature responses are in principle indistinguishable for emission increases from EU, NA and EA, while the SA emission response is weaker in NHml and ARCT while it is stronger in SHext and Tropics compared to the other experiments. The spatial distributions of the temperature responses are shown in Figure 5."*

**L263-L270/Figure 6:**

**I'm confused by the different indications of significance. E.g in Figure 6d, the 10xSA ERF in the tropics has an error bar which does not cross zero, and yet it is shaded to indicate that it's not significant. Yet in Fig 6b, the 5xNA ERF in the NHml has a huge error bar that spans zero, but is shaded to indicate that it is significant.**

We have examined the significance for these results looking at the definition of the student's t-test. Here, the fact that the magnitude of the difference between the 5xNA and control simulation is larger than 10xSA and control simulations is the key factor. The absolute difference between the mean values has a larger influence than the difference in the variances in this case for determining if the result is statistically significant. However, the 10xSA result is close to being significant at the 95% confidence level with p = 0.065.

**L316-318:**

**Consider citing Teng et al. (GRL 2012, doi:10.1029/2012GL051723) which provides a similar example of aerosol forcing over Asia resulting in remote warming over the US, in a different model.**

We thank the reviewer for the reference suggestion. A reference to Teng et al. (2012) has been added, line 543.

*"Moreover, Teng et al. (2012) found a temperature impact in North America directly linked to absorbing aerosols in Asia."*

**L415:**

**Units of climate sensitivity seem to have been inverted here. Check that the number being quoted isn't actually the feedback parameter.**

The number given is indeed a feedback parameter, or climate response parameter, and the value used for the calculations made in the manuscript was the inverted value. The reason for choosing to present the value as a feedback parameter was for transparency with respect to Iversen et al. (2013), Table 1, from where the number is taken. In Iversen et al. (2013), different climate sensitivity estimates are presented as feedback parameters with the units Wm2/K. We noticed, however, that the value had inadvertently been miscited and should read 1.101, not 1.01 Wm2/K. The calculations and figures have been corrected and the conclusions are not affected by this mistake. We have rephrased this paragraph so that it should be clear for the reader that the value cited is the climate sensitivity, line 425.

*"For NorESM this climate sensitivity has been estimated to 0.91 K/Wm −2 (Iversen et al., 2013, λ reg in Table 1)."*

**L422:**

**Why is the goal to use model-dependent sensitivities? Surely for integrated assessment modelling, you would like to use a model-independent choice of climate sensitivity? So then, does it matter if you assume a different climate sensitivity and get a different answer (scaled up or scaled down) as a result?**

The point we would like to make in that paragraph is that there is no model-independent sensitivity or RTP coefficient. We see that this was not clear in the manuscript and have rephrased parts of this paragraph, line 434-439. Even if no climate sensitivity is explicitly applied, there is inherently the climate sensitivity of the model simulations from which the RTP coefficients were derived integrated in the coefficients. The equilibrium sensitivity of the GISS model can only be used by explicitly applying that parameter in the equations for the ARTP. What we wanted to show is how well the RTP coefficients of Shindell and Faluvegi (2010) work without weighting with a climate sensitivity, which means that the sensitivity of that particular model simulation is what is, in effect, used.

If it matters what sensitivity is applied in the equation will depend on what the RTP coefficients will be used for. When comparing the effect of different forcing agents in integrated assessment modelling, it would be crucial, but when comparing the relative latitudinal temperature change distribution, it could be considered less important. However, an assumption about the climate sensitivity always has to be made no matter what the application is, which is what we would like to emphasise. However, an uncertainty related to the RTPs are useful for a sensitivity analysis in the integrated assessment analysis.

**L424:**

**why do you compare with the GISS-E2 transient sensitivity and not equilibrium, given that the NorESM simulations aren't transient?**

The climate sensitivity is an integral part of the formulation of the RTP-coefficients of Shindell and Faluvegi (2010). If a climate sensitivity is not explicitly applied, the implication is that the climate sensitivity from the simulation the RTP coefficients is used. This sensitivity is equal to the transient climate sensitivity of the GISS model (Shindell, 2012). We have rewritten this paragraph to make this clearer for the reader, line 434-439. Please also see discussion above.

*"A third alternative is to apply the RTP coefficients without normalising with a model dependent climate sensitivity parameter, i.e. using the RTP-coefficients of Shindell and Faluvegi (2010), Table 3, directly with forcing estimates (Fig. 14). The implicit assumption in this method is that the sensitivity of NorESM to aerosol forcing is equal to that of the GISS model simulations used to derive the RTP coefficient. This is equivalent with applying the GISS model's sensitivity of 0.5 K/Wm −2 (Shindell, 2012) in Equation 2. This assumption about the sensitivity leads to RTP-derived temperature responses with smaller RMSD values than both those derived by applying the ECS for NorESM in Equation 1 and 2."*

**L427-428:**

**I don't agree with this conclusion. The way I read it, using the Shindell and Faluvegi coefficients has reproduced NorESM well here because GISS-E2 has an ECS (i.e. 2 x CO2) climate sensitivity of 0.6 K/(Wm-2) (Flato et al. IPCC 2013), which happens to be very similar to the sulfate climate sensitivity found for NorESM here. Hansen et al. (JGR 2005, doi:10.1029/2005JD005776) shows that in GISS-E2, the ERF-based sulfate sensitivity is similar to CO2. So this doesn't explain why the authors get such different climate sensitivities for sulfate and 2 x CO2 simulations in NorESM. Maybe due to differences in methodology defining the equilibrium state? Or in calculating ERF (e.g. fixed-SST versus Gregory regression?). At any rate, it would be interesting to understand why NorESM seems to have such a different climate sensitivity for sulfate here compared with the previously published 2 x CO2 values. The message of e.g. the Hansen et al. (2005) paper is that ERF is a more forcing- independent predictor of temperature change, so it's surprising that the global**

**climate sensitivity of NorESM varies so much between forcers. One final point, is that climate sensitivities in general differ hugely between models for the same forcing agent (e.g. the range in Flato et al. is from ~0.5 to 1.5 K/(Wm-2)). This is presumably the case for sulfate as much as any other forcer. So, the coincidence that GISS-E2 has a similar climate sensitivity to NorESM doesn't really show that there is smaller variation across models in the sulfate climate sensitivity compared with between different forcers in the same model; this seems quite unlikely to be the case across most models in fact.**

The point is well taken, this sentence has been removed. We have also clarified that the climate sensitivity derived from the experiments presented here is not directly comparable to the equilibrium climate sensitivity, line 426-428.

*"This is higher than the sensitivity to aerosol forcing obtained in this study. The climate sensitivity from the simulations presented here is not directly comparable with an equilibrium climate sensitivity, since an equilibrium temperature response would require considerably longer simulations for allowing the ocean to fully adjust."*

**L486-487:**

**If saturation of aerosol indirect effects is the explanation here, then shouldn't there be a similar difference in the RF/em as there is in the ERF/em? I don't see how the RF/em can be unaffected by CDNC saturation such that it only shows up as a difference in ERF/em. On a related point: In Figure 4, the error bars for the 0xEU response per em or per RF are enormous and span the entire range of the other experiments. Can the authors be confident that the sensitivity to an emissions reduction actually is any different to an emissions increase, given the considerable overlap of the error bars? It might just be that the smaller forcing and smaller response from the 0x experiment has higher uncertainty because the signal is small compared to internal model variability.**

There is a larger RF response per unit emission for 0xEU compared to 7xEU (Figure 3). In the RF simulations the cloud albedo effect is diagnosed from the cloud distribution determined by the fixed meteorology. Thus, the indirect effect is restricted by the cloud distribution and represent only this first indirect effect (constant cloud water content but changed CDNC). In the ERF simulations the cloud distribution and liquid water content respond to the changed aerosol. In particular, the liquid water path shows large changes which impacts the opacity of the clouds. Thus, including the life-time effect and semidirect effects on clouds amplifies the indirect effects in the ERF simulations compared to the RF simulations.

The results are certainly associated with a lot of uncertainties, as discussed in the manuscript. However, the fact that the results are normalised by the emission change inflates the standard deviation of the 0xEU experiment compared to the other experiments. In fact, the variability is similar in all simulations, se Figure 1 below. The temperature responses in the different experiments have also been tested for significance with the student's t-test, and the global temperature response to European SO2 emission reductions is found to be statistically different to the temperature responses in emission increase experiments, see Figure S1a. We have now clarified this in the manuscript, line 335-336. We have also emphasised in the text that uncertainties are large and that what we present is no proof of nonlinearity, but an indication that there is no reason to assume that the relationship between emission change and temperature response would be linear, line 561-563.

*"The global average temperature change per unit emission in the emission reduction experiment is significantly different from those in the emission increase experiments (Fig. S1)."*

*"Furthermore, indications were found that the emission-based RTPs derived with NorESM might be non-linear. Removal of anthropogenic European SO 2 emissions led to a temperature response per unit emission approximately twice of that in the 7xEU experiment in NorESM. The result is, however, associated with large uncertainties."*

[Figure]

**Technical/grammar/typographic comments:**

**L69-70:**

**Confusing wording in this sentence, please re-phrase**

This sentence has been rephrased, line 70-71.

*"Lately, the radiative forcing of long-lived greenhouse gases other than CO 2 have been included in GAINS, which makes it possible to evaluate the changes emissions of these due to air pollution abatement."*

**L176-177 and Eq3:**

**Inconsistent use of r subscript (emission region or response region?)**

The subscripts have been changed for equation 3 to be consistent with equations 1 and 2.

**L217:**

**add 'typically' or similar caveat**

It has been clarified that this results pertain to the experiments we present in this study, line 224-225.

*"Thus, on a global scale, fast cloud feedbacks contribute to dampen the forcing effect of the emission increases in the NorESM experiments presented here."*

**Figure 4:**

**The caption should explain how the quantities are normalised. Currently, have to refer to the main text to find out that everything is normalised to the 5xNA experiment in this plot.**

We agree with the reviewer. An explanation has been added to the figure caption of Figure 3.

*"Quantities are normalised by the 5xNA response."*

**263:**

**Should Figure 6 have been referenced here?**

We thank the reviewer for noticing this mistake. A reference to Figure 6 has been added here, line 272.

**L299:**

**increase -> increases**

The mistake has been corrected, line 307.

**L495: skills -> skill**

We thank the reviewer for noticing. This has been corrected, line 507.

**Reviewer #2**

**Main comments:**

**1) It is worth citing and mentioning Conley et al. (2018), which looks at climate response to removal of US SO2 emissions. There are some possibilities for comparison and discussion, such as their Table 3 which includes an estimate of temperature response per unit emission change of SO2.**

We thank the reviewer for the suggestion. We have now included a citation of Conley et al. (2018) in the manuscript, line 51 and in the Discussion as well as in Summary and Conclusions.

*"…and how these processes are represented in different climate models (Kasoar et al., 2016; Conley et al., 2018)."*

**2) Global maps of the temperature response to each of these SO2 perturbations would strengthen this paper. Likewise I think global maps of ERF would be interesting as well. This would put the results in context of the few multimodel studies on this topic, such as the one cited above and Kasoar et al. (2016) which you mention already in the manuscript.**

As advised by both reviewer #1 and #2, global maps of the temperature response and the ERF are now added to the manuscript, please see Figure 5 and 7.

**3) The biggest weakness of the paper is the use of a single coupled climate model, especially in a time when multimodel studies are becoming the norm. Ideally, the emissions-based RTP coefficients could be based on an average of several disparate models for more robustness. I understand that it's not feasible to do that in this study, but perhaps the authors could comment on whether or not they expect their results to be robust across additional CMIP models?**

We have added a comparison with the results of Conley et al. (2018) and Kasoar et al. (2018), and we do find similarities between the results. The strong Arctic response is present in these two studies as well, line 533-536.

*"These results are in line with those of Conley et al. (2018), who found a similar latitudinal temperature change distribution in three different models in response to removal of US SO 2 emissions, and Kasoar et al. (2018) who conducted a single model study where they found that the Arctic warmed most in response to removal of SO2 emissions in different regions."*

**4) In Fig 4a and b, the error bars for just one standard deviation from the mean are quite large for the zero EU SO2 emissions perturbation How can the authors then be so sure about a nonlinearity in the response depending on the magnitude and sign of the emissions changes? Since the zero-out EU SO2 emissions perturbation is much smaller in absolute magnitude than the 7xEU, you would likely need a slightly longer simulation than 160 years to reduce those error bars. Otherwise, I'm not sure how you can rule out the role of internal climate variability.**

Indeed, the 0xEU simulation (and control simulation) is 200 years (line 165). However, the magnitude of the variability is not larger in the 0xEU simulation compared to the other simulations (please also cf. answer to reviewer #1 and Figure 1 above). The non-normalised variability is similar in all simulations. We see that the way of presenting the result as normalised quantities obscures this fact and in the revised manuscript we use the standard error instead of the standard deviation as an indication of uncertainty, line 166-167, and emphasised that the result is associated with uncertainties, line 561-563.

*"Uncertainty ranges for the results are given as standard errors or standard deviations derived from the variability in each simulation."*

*"Furthermore, indications were found that the emission-based RTPs derived with NorESM might be non-linear. Removal of anthropogenic European SO 2 emissions led to a temperature response per unit emission approximately twice of that in the 7xEU experiment in NorESM. The result is, however, associated with large uncertainties."*

**Minor comments**

**1) I'm not really seeing the grey shading in Figure 5? Is it there but just really small?**

The shading colour in what previously was Figure 5, now Figure 4, has been adjusted to appear more grey. Note that it is only the SHext bars in Figure 4 that are shaded grey.

**2) L332-333: this isn't a complete sentence. In general I think the phrase "e.g." is overused in this manuscript and seems to be rather unconventional to start sentences with that abbreviation which happens a couple of times here.**

E.g. has been changed to "For example", line 342. We have gone through the manuscript with particular attention to this phrase and exchanged with equivalent phrase to make the text more varied.

**3) L84: "an comparison" should be "a comparison"**

"An" has been changed to "a", line 86. We thank the reviewer for noticing the mistake.

**References**

[revised manuscript text omitted]